# Medial prefrontal cortex and anteromedial thalamus interaction regulates goal-directed behavior and dopaminergic neuron activity

Chen Yang [1,2,7], Yuzheng Hu [3,4,7], Aleksandr D. Talishinsky [1], Christian T. Potter [1], Coleman B. Calva[1], Leslie A. Ramsey [5], Andrew J. Kesner [1], Reuben F. Don[1], Sue Junn[1], Aaron Tan[1], Anne F. Pierce[1], Céline Nicolas [1], Yosuke Arima[1], Seung-Chan Lee[1], Conghui Su [3], Jensine M. Coudriet[1], Carlos A. Mejia-Aponte[6], Dong V. Wang [1], Hanbing Lu[4], Yihong Yang[4] & Satoshi Ikemoto [1]✉

The prefrontal cortex is involved in goal-directed behavior. Here, we investigate circuits of the PFC regulating motivation, reinforcement, and its relationship to dopamine neuron activity. Stimulation of medial PFC (mPFC) neurons in mice activated many downstream regions, as shown by fMRI. Axonal terminal stimulation of mPFC neurons in downstream regions, including the anteromedial thalamic nucleus (AM), reinforced behavior and activated mid-brain dopaminergic neurons. The stimulation of AM neurons projecting to the mPFC also reinforced behavior and activated dopamine neurons, and mPFC and AM showed a positive-feedback loop organization. We also found using fMRI in human participants watching reinforcing video clips that there is reciprocal excitatory functional connectivity, as well as co-activation of the two regions. Our results suggest that this cortico-thalamic loop regulates motivation, reinforcement, and dopaminergic neuron activity.

[1] Neurocircuitry of Motivation Section, National Institute on Drug Abuse, National Institutes of Health, Baltimore, MD 21224, USA. [2] Department of Neurosurgery, Tangdu Hospital, Fourth Military Medical University, No.1, Xinsi Road, 710038 Xi'an, Shaanxi, P. R. China. [3] Department of Psychology and Behavioral Sciences, Zhejiang University, 310028 Hangzhou, P. R. China. [4] MR Imaging and Spectroscopy Section, National Institute on Drug Abuse, National Institutes of Health, Baltimore, MD 21224, USA. [5] Ex Vivo Electrophysiology Core, National Institute on Drug Abuse, National Institutes of Health, Baltimore, MD 21224, USA. [6] Histology Core, National Institute on Drug Abuse, National Institutes of Health, Baltimore, MD 21224, USA. [7] These authors contributed equally: Chen Yang, Yuzheng Hu. ✉email: satoshi.ikemoto@nih.gov

The prefrontal cortex (PFC) has been implicated in executive control of goal-directed behavior[1–4]. Consistently, dysregulation of the PFC is associated with disorders of goal-directed behavior, including drug addiction and depression[5–8]. Recently, these disorders have been treated with PFC stimulation procedures such as deep brain stimulation (DBS) and transcranial magnetic stimulation (TMS), with promising results[9–13]. The effects of such treatments most likely depend on large-scale brain circuitry regulating the motivation that drives goal-directed behavior[14], and the details of such circuitry are currently incomplete.

Goal-directed behavior emerges when behavior is reinforced by stimuli/events and can be considered with respect to three key aspects: its motor function (i.e., coordination of movements), cognitive function (i.e., understanding of the external environment), and the motivation function (i.e., induced arousal that drives attention and action to a goal)[15]. The PFC is thought to regulate these aspects of goal-directed behavior through interactions with the basal ganglia[16]. The medial structures of the basal ganglia, including the ventral striatum (VStr) and the ventral tegmental area (VTA), which receive inputs from the medial PFC (mPFC), are thought to be important in the motivational aspect for goal-directed behavior (hereafter referred to as goal-directed motivation)[17–20]. Consistently, mPFC stimulation can activate the VTA-VStr dopaminergic (DA) neurons[17,21–23], whose dysregulation has been implicated in mood and substance-use disorders[24,25]. Although accumulating evidence strongly supports the claim that the mPFC interacts with the VStr and VTA for goal-directed motivation, the mPFC's projections to extra-basal ganglia regions may also play a role. While the thalamus has been implicated in goal-directed behavior, based primarily on connectivity findings[26], direct evidence has not yet been provided to support that mPFC projection to the thalamus play a role in goal-directed motivation.

The aim of this study is to systematically investigate the PFC's neuroarchitecture in goal-directed motivation. We measured goal-directed motivation using the rate of intracranial self-stimulation behavior (ICSS) in which mice are given the opportunity to respond for optogenetic stimulation of neural populations at cell bodies or terminals. First, we systematically compared PFC areas for ICSS rates, to test which PFC areas are more important in goal-directed motivation[16,27,28], and found that the mPFC supports the highest rates of ICSS among PFC areas. Second, we performed functional MRI (fMRI) with mPFC stimulation, revealing increased activation in various downstream regions including the VStr and extra-basal ganglia regions such as the thalamus and hypothalamus. Third, we found that ICSS was triggered by terminal stimulation of mPFC neurons at various downstream regions, including the anteromedial thalamic nucleus (AM). Fourth, we found that the stimulation of both AM neurons and the AM-to-mPFC pathway supports ICSS and that the AM organizes a positive-feedback loop with the mPFC. Fifth, we found that the stimulation of both mPFC-to-AM and AM-to-mPFC neurons activated VTA DA neurons and that the inhibition of VTA DA neurons decreased the rates of ICSS reinforced by the stimulation of these pathways. Finally, we conducted a proof-of-concept experiment providing evidence for the co-activation of mPFC and AM in humans during motivated behavior. In summary, this study found that the PFC interacts with extra-basal ganglia regions in goal-directed motivation and identified a cortico-thalamic positive-feedback loop that can regulate VTA DA neurons and goal-directed motivation.

## Results

**mPFC neurons support high rates of ICSS.** The PFC can be divided into three major areas based on connectivity[29,30] (Supplementary Note 1): the mPFC (the infralimbic, prelimbic, cingulate area 2, and medial and ventral orbital regions), orbital PFC (the lateral orbital and anterior insular regions), and motor PFC (the cingulate area 1 and premotor regions). These areas are thought to be important in the motivational, cognitive, and motor aspects of goal-directed behavior[16,31], respectively. However, it has not yet been documented whether the mPFC is involved in goal-directed motivation more closely than those of the other PFC areas are. To this end, we systematically compared the effects of optogenetic ICSS across areas of PFC using the ICSS rate as the measure of goal-directed motivation. In the interest of precision, we initially distinguished the ventral mPFC or the infralimbic region (IL) from the rest of the mPFC, because previous studies often suggested the distinguished role of the IL in motivation and DA activity (e.g., refs. [32–35]). As an anatomical control, we also examined the tenia tecta (TT), which is positioned immediately ventral to the IL and part of the olfactory system that may be involved in goal-directed motivation[36].

Each mouse received an injection of adeno-associated viral vector (AAV) with human synapsin promoter (hSyn) carrying channelrhodopsin-2 (ChR2) and enhanced yellow fluorescent protein (EYFP) into one of the areas and had an optic fiber implanted for subsequent photostimulation (Fig. 1a–c). Some mice received the control vector AAV-hSyn-EYFP into the IL. Three weeks after the surgery, each experimentally naive mouse was placed in an operant conditioning chamber, equipped with two levers (Fig. 1d). Responding on the left "active" lever was rewarded with a train of photostimulation (25 Hz, 8 pulses) in sessions 3–7 (the acquisition phase) and 11–14 (re-acquisition), while no photostimulation was delivered in sessions 1–2 (baseline) or 8–10 (extinction). Responding on the right "inactive" lever produced no consequence throughout sessions 1–14. The mice with IL stimulation markedly increased responding on the active lever in sessions 3–7, decreased lever pressing in sessions 8–10, and reinstated lever pressing in sessions 11–14 (Fig. 1e; mPFC: IL: Active Lever; Supplementary Fig. 1). Response levels on the inactive lever did not increase or change throughout the experiment (Fig. 1e; mPFC: IL: Inactive Lever). Control mice, which received IL stimulation, but expressed no ChR2, did not increase lever pressing over baseline levels (mPFC: IL: Control). The data of the control mice established active-lever response levels that determined self-stimulator (>64) from non-self-stimulator, a criterion that is used to interpret ICSS data (mean lever presses plus three times their standard deviation; see "Methods"). Photostimulation at other mPFC regions supported ICSS at rates similar to those of IL stimulation (Fig. 1e; mPFC: PrL + Cg2 + mOrb + vOrb; Supplementary Fig. 1; a $2^{sub-area}$ x $5^{session}$ mixed ANOVA on lever presses: no main area effect, $F_{1,15} = 0.012$, $P = 0.92$). Because the stimulation of the IL supported quantitatively similar levels of ICSS as that of the other mPFC sub-regions, these sub-regions of the mPFC are grouped for the rest of the study.

Stimulation of the orbital PFC produced inconsistent effects: It supported ICSS in some mice with moderate rates, while did not in others (Fig. 1e; orbital PFC). By contrast, photostimulation of the motor PFC did not support ICSS (Fig. 1e; motor PFC). Interestingly, stimulation of the TT reliably supported ICSS, though at low rates (Fig. 1e; TT). The five groups of mPFC, orbital PFC, motor PFC, TT, and control mice did not differ in ICSS rates during the first two sessions during which no photostimulation was available (a $5^{group} \times 2^{session}$ mixed ANOVA on lever presses: $F_{4,45} = 1.86$, $P = 0.135$); however, during the acquisition phase (sessions 3–7), the mPFC supported ICSS at rates significantly greater than the other groups ($P = 0.0002$, a post hoc Tukey HSD test after a $5^{group} \times 5^{session}$ mixed ANOVA on lever presses: a significant main area effect, $F_{4,45} = 18.48$, $P < 0.0001$). These results support the idea that the mPFC

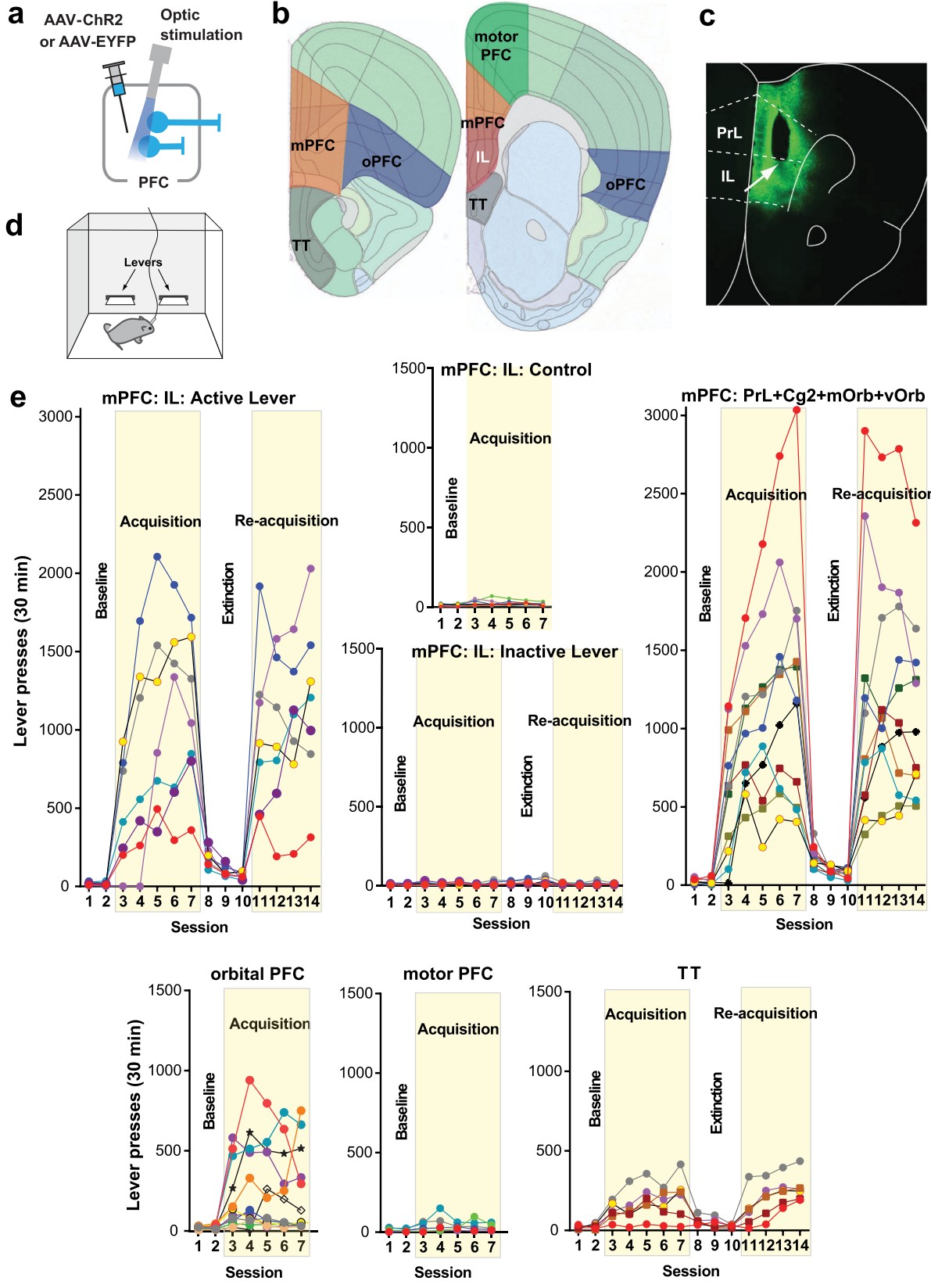

regulates goal-directed motivation more than other PFC areas. All subsequent experiments focused on the mPFC.

**mPFC-induced motivation is short-lasting and depends on neural firing frequency**. The mice with mPFC stimulation

performed ICSS with a median response of 1180 in session 7 (Fig. 1e). That is, half of the mice responded on the lever once every 1.5 s or greater over the course of the 30-min session, and many mPFC mice displayed lever pressing at a constant, fast rate after a few sessions (e.g., Fig. 2a). The highest responder performed at the rate of 1.7 presses every second for a 30-min

**Fig. 1 mPFC neurons support high rates of ICSS. a** C57BL6 mice received an injection of either AAV-hSyn-ChR2-EYFP or AAV-hSyn-EYFP into one of the cortical regions and an optic-fiber implant. **b** Five injection areas. IL infralimbic region, mPFC medial prefrontal cortex, oPFC orbital PFC, PrL prelimbic region, TT tenia tecta. **c** Coronal sections show a representative EYFP expression with the tip of an optic-fiber placement (the arrow) just above the IL. Coronal sections were adopted from Allen Mouse Brain Atlas© of Allen Institute for Brain Science available at: https://mouse.brain-map.org/static/atlas. **d** Schematic showing a mouse in a conditioning chamber equipped with two levers for ICSS. **e** ICSS data: Responding on the left lever was rewarded with photostimulation (a 25-Hz 8-pulse train) in sessions 3–7 and 11–14, while no photostimulation in sessions 1–2 or 8–10. Each line shows lever-press data from a single mouse. mOrb medial orbital region, Cg2 cingulate area 2, vOrb ventral orbital region.

---

session, suggesting that mPFC neurons can drive vigorous motivation under optimal conditions (depending on probe placement, ChR2 expression level, etc.). In addition, when reinforcement contingency was reversed between the two levers, the mice quickly learned to reverse response choice from the formerly active lever to the currently active lever within a session, a significant lever-by-session interaction by a $2^{lever} \times 3^{session}$ ANOVA, $F_{2,16} = 36.8$, $P < 0.0001$ (Fig. 2b, c), suggesting that the responding was controlled by induced motivation (i.e., goal-directed emotional arousal) rather than habit formation resulting from repeated fixed-action reinforcement. Moreover, responding became erratic when the reinforcement schedule was strained: When two or more responses are required for a single train of mPFC stimulation, which introduces a temporal gap between stimulation, the mice markedly decreased lever-pressing rates (a $3^{ratio} \times 3^{session}$ ANOVA: the main ratio effect, $F_{2,16} = 19.0$, $P < 0.0001$; no ratio-by-session interaction, $F_{4,32} = 0.15$, $P = 0.96$), and repeating the procedure did not improve their performance (Fig. 2d). A similar effect was also found with increases in reinforcement intervals, a result described in the fMRI experiment below. Although these observations can be explained by increased physical or cognitive effort, we believe that this is consistent with the idea that stimulation makes the animal seek the next stimulation and that such stimulation-induced motivation is short-lasting. This idea is supported by the next experiment where the effect on ICSS rates diminished with increased photo-pulse interval.

We examined how the rate of ICSS changes as a function of stimulation-train frequency (i.e., photo-pulse interval), to determine how important it is for train pulses to occur in a temporally close manner for motivation. While keeping physical (i.e., lever-pressing action) and cognitive (i.e., knowledge of lever-stimulation contingency) demands and reward quantity (i.e., the number of pulses: 8) constant, the mice received trains with varied pulse-intervals (Fig. 2e). That is, while the amount of reward delivered per response was constant, the speed at which fragments of reward was delivered varied. We found that the mice had significantly higher rates of ICSS for trains with 25 or 50 Hz than 6 or 13 Hz (Fig. 2f, bar graph). Active lever-press counts occurring during the second 5-min period of each 10-min block were analyzed with a $2^{order} \times 4^{frequency}$ within-subjects ANOVA. The main frequency effect was highly significant ($F_{3,24} = 41.10$, $P < 0.0001$). Although the order-by-frequency interaction was significant ($F_{3,24} = 5.03$, $P = 0.0076$), the effect was small, and the main order effect was not ($F_{1,8} = 0.11$, $P = 0.747$); therefore, the ascending- and descending-order data are combined and presented irrespective of the order factor. The 50-Hz stimulation was significantly more effective than the 25-Hz stimulation, while the 25-Hz stimulation was significantly more effective than the 13-Hz stimulation, which was more effective than the 6-Hz stimulation. However, it was expected for the mice to obtain trains at lower rates for low-frequency trains, because low-frequency trains take longer to deliver than high-frequency trains. Therefore, we calculated how the mice would have responded if they responded to 6, 13, and 25-Hz trains at the same interval as 50-Hz train (Fig. 2f, green line graph). We found significant differences between expected lever-presses and actual lever-

presses for 6- or 13-Hz trains (cf. the bar and line graphs of Fig. 2f). Therefore, even though the reward amount and effort were constant, the pulse interval determined ICSS rates, suggesting that motivational effect of each pulse decays rapidly within the order of tens of milliseconds (see Supplementary Fig. 2 for conceptual models). Using an opto-tetrode probe for simultaneous stimulation and recording, we confirmed that each train can trigger action potentials consistent with its frequency (Fig. 2g, h), suggesting that the frequency of optogenetic stimulation can regulate the firing rate of mPFC neurons. Therefore, the greater firing rates of mPFC neurons, the greater goal-directed motivation.

We conducted additional experiments to further characterize the contribution of mPFC neurons to motivated behavior. First, inhibiting mPFC neurons with the opsin enhanced halorhodopsin 3.0 (NpHR) did not alter preference for environment paired with laser, motor performance, or exploration in an open field, although it effectively reduced the number c-Fos positive neurons (Supplementary Fig. 3), suggesting that baseline activity of mPFC neurons does not appear to be involved in regulating affective valance. In addition, we examined the contributions of mPFC glutamatergic principal neurons and GABAergic interneurons in ICSS, using transgenic mice with AAV-ChR2-EYFP in a Cre-recombinase-dependent double-inverted open reading frame (DIO) construct. Goal-directed behavior was supported by activation of glutamatergic neurons, but not by GABA neurons (Supplementary Fig. 4).

**mPFC stimulation increases activation in downstream regions, including the basal ganglia and others.** We performed an fMRI experiment to identify regions downstream of mPFC that are recruited during stimulation that supported ICSS. Rats were used for this experiment because the rat brain provides blood oxygenation level-dependent (BOLD) signals with greater resolution and because it is more difficult to perform fMRI in smaller mouse brains[37], which are more prone to imaging artifacts. Rats received a unilateral injection of AAV-hSyn-ChR2-EYFP or AAV-hSyn-EYFP into the mPFC followed by the implantation of an optic fiber above the injection site (Fig. 3a). Rats quickly learned to self-stimulate with mPFC photostimulation (25-Hz 8-pulse train), and then we examined effects of reinforcement intervals: 0, 1, 2, and 4 s, during which the rats could not obtain the next mPFC stimulation upon pressing the lever. Similar to the mice data (Fig. 2d, f), we found that ICSS rates markedly decreased when responding was not reinforced by mPFC stimulation without interruption, in this case increased reinforcement intervals ($F_{3,18} = 3.70$, $P = 0.031$, a $2^{order} \times 4^{interval}$ within-subjects ANOVA; Fig. 3b), a result consistent with the notion that stimulation-induced goal-directed motivation diminishes quickly within a second (Supplementary Fig. 2). It should be noted that although all rats did learn to self-stimulate with mPFC stimulation, the rates of the rats' ICSS were generally lower than those of the mice. Because of the larger brain size of rats, a greater photostimulation power or multiple probes may have been needed to effect the similar number of neurons as in mice, leading rats to achieve greater ICSS rates.

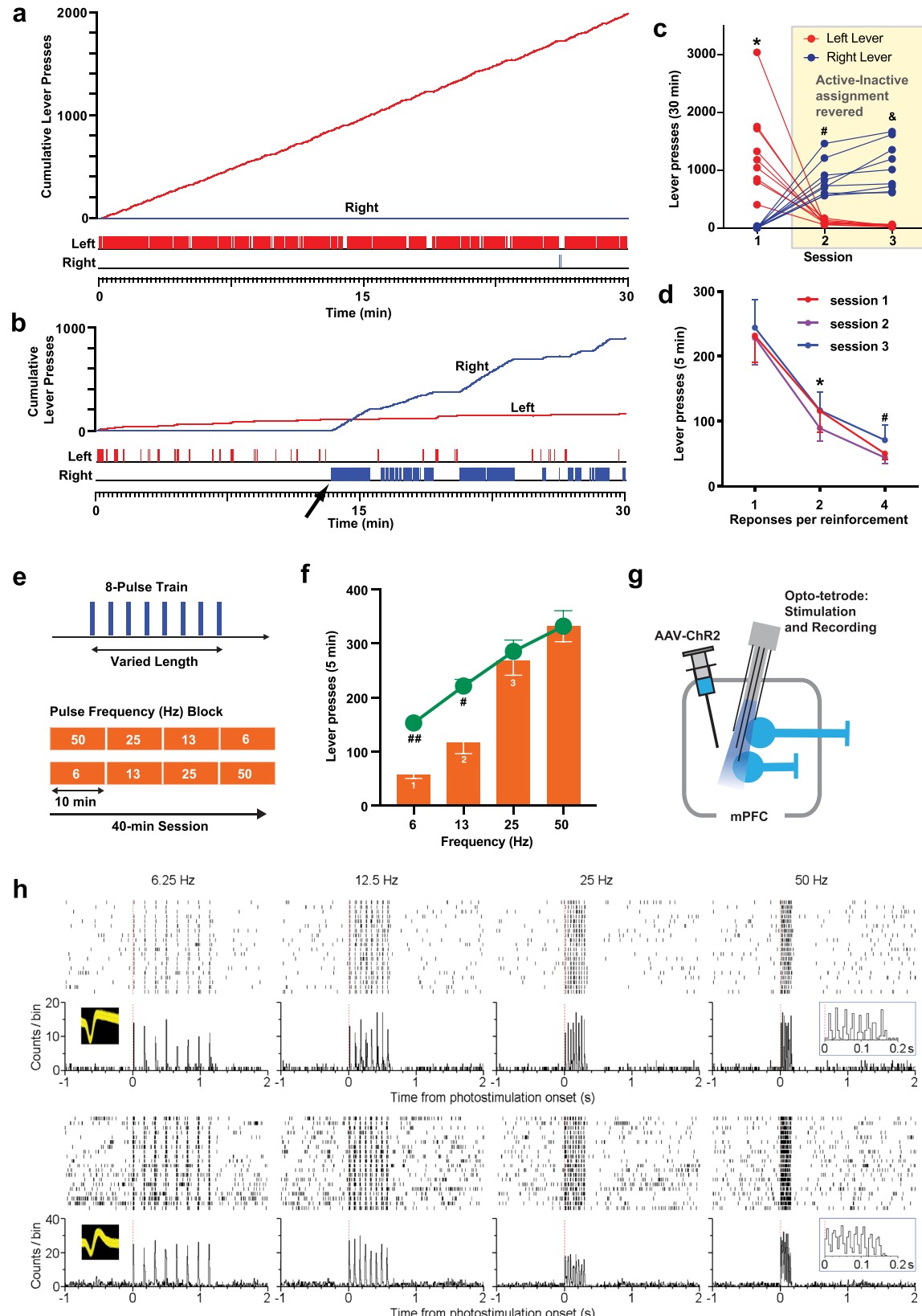

fMRI performed in these rats under anesthesia showed that the interval of unilateral mPFC stimulation determined the strength of BOLD signals (Fig. 3c, d and Supplementary Fig. 5), demonstrating that neural activation elicited by mPFC stimulation decays quickly. The mPFC stimulation increased BOLD signals bilaterally in the entire mPFC, including the IL, prelimbic,

medial orbital, and cingulate area 2 regions, an effect consistent with extensive reciprocal connectivity among mPFC neurons within and between the hemispheres[30] and ICSS data (Fig. 1e and Supplementary Fig. 1). In addition, mPFC stimulation significantly increased BOLD signals in other cortical and subcortical regions including the anterior insular area, tenia tecta, amygdala,

**Fig. 2 mPFC-induced motivation is short-lasting and depends on neural firing frequency. a**, **b** Example of the cumulative record (top) and event record (bottom) showing lever-press responses of a mouse. **b** The cumulative record (top) and event record (bottom) of the mouse (**a**) that was subjected to reinforcement-contingency reversal where the assignment of active and inactive levers were reversed between the left and right levers for the first time. **c** The left and right lever-presses of individual mice ($N = 9$) before and after reinforcement-contingency reversal between the two levers. *Greater than right lever value in session 1 ($P < 0.0001$), left lever value in session 2 ($P = 0.0003$), and left lever value in session 3 ($P < 0.0001$); #greater than right lever value on session 1 ($P < 0.0001$) and left lever value in session 2 ($P = 0.0018$); &greater than right lever value on session 1 ($P < 0.0001$) and left lever value in session 3 ($P < 0.0001$; Tukey HSD post hoc test). **d** Mean lever-presses (±SEM; $N = 9$) decreased markedly when the mice had to lever-press two or four times for a mPFC stimulation. *lower than the ratio-1 value ($P < 0.0011$); #lower than the ratio-1 value ($P < 0.0002$; Tukey HSD post hoc test). **e** ICSS was examined over two 40-min sessions where frequencies (6, 13, 25, and 50 Hz) of the train (top: 8-pulse trains) were changed every 10 min (or two 5-min bins) in an ascending and descending order (bottom); the number of presses during the second 5-min bin was used as the measure of frequency effect. **f** Orange bars (means ± SEM) show effects of frequency on response rates. (1) Significantly different from the 13-Hz ($P = 0.0195$), 25-Hz ($P < 0.0001$), and 50-Hz ($P < 0.0001$) values; (2) significantly different from the 25-Hz ($P = 0.005$) and 50-Hz ($P = 0.0005$) values; (3) significantly different from the 50-Hz value ($P = 0.0035$; Tukey HSD post hoc test). Green line graph (means ± SEM) shows hypothetical response rates if response rates of 6, 13, and 25-Hz trains were self-administered at the same interval (or motivational force) as 50-Hz trains. #$P = 0.001$ ($t_{18} = 4.23$), ##$P < 0.0001$ ($t_{18} = 9.81$), significantly different from the actual responses (unpaired, two-tailed $t$ tests with Benjamini and Hochberg correction). **g** An opto-tetrode probe combining an optic fiber with four tetrodes was surgically implanted for subsequent mPFC stimulation and recording in freely moving mice. **h** Activities of a putative glutamatergic neuron (top) and a putative GABAergic interneuron (bottom). A close relationship between stimulation frequencies and firing rates was confirmed: the greater stimulation frequencies, the greater firing rates.

VStr, the medial part of the dorsal striatum (mDStr), internal capsule (IC), septal area, bed nucleus of stria terminalis, anterior thalamic area, preoptic area, and lateral hypothalamic area. While stimulation-evoked BOLD signals of the mPFC, septum, preoptic area, and amygdala were not significantly correlated with ICSS levels, BOLD signals of the anterior insular area, VStr, anterior thalamus, and hypothalamus were significantly correlated with ICSS levels (Fig. 3e), raising the possibility that these regions integrate goal-directed motivation signals elicited by mPFC stimulation. It should be noted that the lack of correlation between BOLD changes in the stimulation site (i.e., mPFC) and lever presses may be explained by the ceiling effect of BOLD changes, which failed to reflect the real extent of neural activation. Also, robust stimulation-evoked BOLD signals found in the visual thalamus and superior colliculus are readily explained by the stimulation of retinal photoreceptors because similar activations were also found in the control group (Fig. 3d and Supplementary Fig. 5); retinal photoreceptors must have detected light emitted from the external optical cable since testing was conducted in complete darkness.

**Axonal terminal stimulation of mPFC neurons supports ICSS in many downstream regions, including the anteromedial thalamic nucleus.** Guided by results from our fMRI experiment, we performed ICSS experiments with stimulation in mPFC terminals in downstream regions, to determine the extent to which these pathways are indeed involved in goal-directed motivation. Experimentally naïve C57BL/6J mice received an AAV-hSyn-ChR2 injection into the mPFC and an optic-fiber implantation at one of the downstream regions (Fig. 4a, b). They quickly learned to respond on the active lever that delivered the stimulation of mPFC terminals at subcortical regions including the VStr. This observation is consistent with the established role of mPFC-to-VStr neurons in goal-directed behavior[7,8,16,19]. In addition to the VStr, we examined the mDStr and IC in the striatum (Fig. 4c, d) because mPFC stimulation increased BOLD signals in the mDStr and IC. While it may not be well acknowledged, mPFC projects to the mDStr[38–40] and the IC through which mPFC neurons reach many downstream regions, including the thalamus and hypothalamus[41]. Indeed, the stimulation of mPFC terminals at the mDStr and IC supported strong ICSS. We further dissected the mPFC-IC pathway by placing fibers more precisely in several mPFC terminal fields and found that the septum did not support ICSS. For hypothalamic stimulation, stimulation sites were placed in the lateral part, since the

stimulation of the lateral, but not medial, hypothalamic area supports vigorous ICSS[42]; however, all sites tested in the hypothalamus supported relatively moderate ICSS. Of thalamic nuclei, the mediodorsal thalamic nucleus (MD), which is reciprocally linked with the PFC, supported relatively modest ICSS. Similarly, the reuniens thalamic nucleus (Re), which receives mPFC afferents, either supported modest ICSS or none at all. However, we found high rates of ICSS with the stimulation of mPFC terminals in the anterior thalamic area (ATh), particularly the anteromedial thalamic nucleus (AM).

**mPFC neurons projecting to the AM have collateral projections to the VStr and VTA.** It is unknown whether and how the AM is involved in motivated behavior. One possibility is that the stimulation of mPFC terminals in the AM activates the collaterals of the same mPFC neurons projecting to the VTA or VStr; therefore, AM neurons may not be involved in the ICSS effects per se. To shed light on this issue, we first examined possible collaterals using an AAV-FLEX-mGFP-2A-SYP-mRuby, which fills the entire Cre-containing cells with mGFP and their terminal boutons with mRuby. We injected this vector into the mPFC and AAV-retro-eGFP-Cre into the AM in C57BL/6J mice ($n = 3$; Fig. 5 and Supplementary Figs. 6 and 7). This procedure should label the axons of mPFC neurons projecting to the VTA or VStr with mGFP and the terminals with mRuby, if mPFC neurons projecting to the AM have collaterals to these regions. mRuby-labeled terminals were found confined in the AM with no detectable diffusion to the MD or Re (Fig. 5f and Supplementary Figs. 6f and 7f) and resulted in mRuby labeling of terminal boutons in the VStr (panels d, g), and VTA (panels e, h). In general, the collaterals of mPFC–AM neurons to the VStr were more robust than those to the VTA (cf. panels f and g of Fig. 5 and Supplementary Figs. 6 and 7). These results suggest that mPFC neurons projecting to the AM have collateral projections to the VStr, VTA, or both, implying that mPFC neurons coordinate the activities of these regions.

**Stimulation of AM neurons supports ICSS.** We next examined whether direct stimulation of AM neurons supports ICSS. Experimentally naïve C57BL/6J mice received an AAV-hSyn-ChR2 injection into the AM and an optic-fiber implantation at the AM ($n = 5$; Fig. 6a and Supplementary Fig. 8). They quickly learned to respond on the active lever that stimulated AM neurons. Because AM neurons primarily project back to the

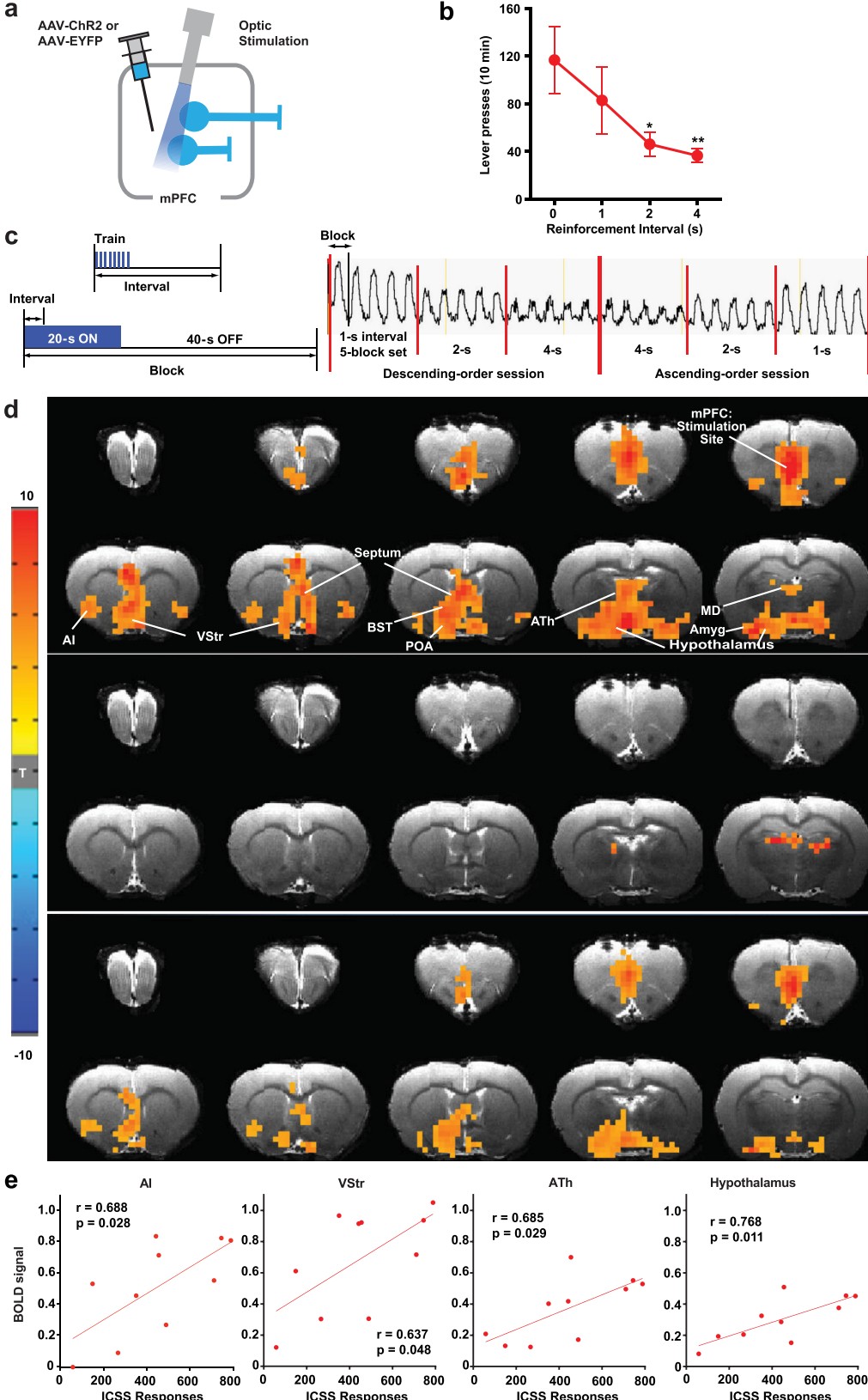

mPFC[43–45], we then examined whether stimulation of AM terminals in the mPFC supports ICSS. Experimentally naive C57BL/6J mice received an AAV-hSyn-ChR2 injected into the AM and an optic fiber implanted in the mPFC ($n = 5$; Fig. 6b). Mice quickly learn to respond on the active lever that delivered photostimulation, suggesting that the stimulation of the AM-to-

mPFC pathway supports ICSS. Because it is difficult to infect AM neurons without affecting nearby neurons that also project to the mPFC with the microinjection of AAV into the AM, we also examined the same question with a retrograde serotype of an AAV-ChR2 injected into the mPFC and an optic fiber implanted in the AM ($n = 7$; Fig. 6c). The mice quickly learned to respond

**Fig. 3 mPFC stimulation increases fMRI BOLD signals in downstream regions. a** Each rat received AAV-ChR2-EYFP or AAV-EYFP injected into the IL followed by the implantation of an optic fiber at the same region. **b** ICSS: The rats ($n = 7$) received photostimulation (a train of 8 pulses at 25 Hz) upon lever pressing. After a stable response over a few sessions, the effects of the reinforcement interval (0, 1, 2, and 4 s) were evaluated with a within-subjects design in that order. Each interval was tested in a 10-min block. Data are mean lever-presses (±SEM). *$P = 0.0319$, **$P = 0.0135$, significantly lower than the continuous-reinforcement schedule (0 s) value (Tukey HSD post hoc test). **c** fMRI: Photostimulation (a train of 8 pulses at 25 Hz) delivered at the IL with one of three intervals (1, 2, or 4 s) during the 20-s ON phase of a 60-s block. Each block was repeated 5 times for each train interval. Effects of reinforcement intervals were determined in both descending- and ascending-order sessions. The vertical axis indicates raw BOLD signal intensity of one mPFC voxel in arbitrary unit. **d** The unilateral photostimulation with the 1-s interval activated the entire medial network of the PFC, including the medial orbital, prelimbic, ventral anterior cingulate, and infralimbic regions, with stimulation-side dominance. It also elicited significant BOLD signals in extensive subcortical regions of the mPFC (top; $n = 10$), but limited signals in the control group (middle; $n = 7$). Significant differences in BOLD signal between the ChR2 and control groups are shown in the bottom panel. Imaging results were corrected for whole-brain multiple comparisons (Corrected $P < 0.05$). AI anterior insular region, Amyg amygdala, ATh anterior thalamic area, BST bed nucleus of stria terminalis, MD mediodorsal region, POA preoptic area, VP ventral pallidum, VStr ventral striatum. **e** Significant correlations (Pearson's *r*) between ICSS responses and the levels of BOLD signals in the AI, VStr, ATh, and hypothalamus. Each dot represents the data of a single rat.

for the stimulation and displayed robust ICSS levels. We verified that each stimulation probe was found within the AM (Fig. 6c, right and Supplementary Fig. 8c). The areas that stimulated tended to have low fluorophore expression levels presumably due to photobleaching. These results suggest that the stimulation of AM-to-mPFC neurons supports ICSS.

**AM neurons form a positive-feedback loop with mPFC neurons.** Many known pathways involved in goal-directed motivation have reciprocal connectivity and function via positive feedback to promote motivated behavior[18]. To provide evidence for a positive-feedback organization between the mPFC and the AM, we conducted brain slice electrophysiology experiments. Specifically, we examined whether the stimulation of mPFC-to-AM neurons excites AM-to-mPFC neurons and vice versa (Fig. 6d, e). We expressed ChR2 in mPFC neurons by injecting AAV-EF1α-DIO-ChR2-EYFP and the retrograde tracer AAV-retro-tdTomato into the mPFC of Emx1-Cre mice, to prevent possible retrograde infection to AM neurons. Emx1 is selectively expressed in cortical neurons, but not thalamic neurons[46]. Of the ten cells recorded in the AM, nine were light-responsive (light-evoked current $254 \pm 82$ pA (mean ± SEM). The application of glutamate receptor antagonists induced a nearly complete blockade of evoked EPSC in all tested AM neurons ($n = 4$, Fig. 6d, f). For EPSC evoked in mPFC by the AM-to-mPFC pathway, C57BL/6J mice received an injection containing AAV-hSyn-ChR2-EYFP and the retrograde tracer AAV-retro-tdTomato into the AM (Fig. 6e). Of the eight cells recorded in mPFC, eight were light-responsive, confirming that AM neurons can excite mPFC-to-AM neurons (Fig. 6e, f). Bath application of glutamate receptor antagonists resulted in a 62% inhibition of EPSC of the mPFC-to-AM pathway. EPSC latency of AM neurons was $3.0$ ms ± $0.13$ ms (mean ± SEM), while that of mPFC neurons was $5.1$ ms ± $0.51$ ms, and mPFC responded significantly slower than AM neurons (Fig. 6g). Together with the ICSS data, these data suggest a positive-feedback organization between the mPFC and the AM in goal-directed motivation.

**mPFC-to-AM neurons or AM-to-mPFC neurons regulate the activity of VTA DA neurons.** VTA DA neurons play a key role in motivation and reinforcement, and the mPFC can regulate VTA DA neurons. Microdialysis studies have shown that electrical stimulation of the mPFC activates the mesolimbic DA system[21–23], and recently, optogenetic stimulation of the anterior forebrain projecting to the VTA was shown to excite DA neurons[17]. Because this study found a role of the mPFC-to-AM and AM-to-mPFC pathways in ICSS, we used a fiber-photometry calcium-signaling procedure in TH-Cre mice to examine whether

the stimulation of the mPFC-to-AM or AM-to-mPFC pathway activates VTA DA neurons. Due to its role in motivation and reinforcement[19,47,48], the mPFC-to-VStr pathway was also examined in the same mice (Fig. 7a). We found that the stimulation of both the mPFC-to-AM pathway and the mPFC-to-VStr pathway significantly increased GCaMP signals in the VTA (Fig. 7b; see Supplementary Figs. 9 and 10 for detailed data and statistical analyses). The strength of the recorded signals depended on several factors coming together: the locations of the tips of stimulation and recording probes in relation to the target regions and expression levels opsins at the target regions (e.g., MNV3 and 4 of Fig. 7b). In other words, when one of these conditions was compromised, signals were compromised. In addition, we found that the stimulation of the AM-to-mPFC pathway significantly increased GCaMP signals in the VTA (Fig. 7c, d and Supplementary Figs. 9 and 10).

While previous studies showed that mPFC ICSS depends on DA transmission[49–52], it is unclear whether the ICSS reinforced by the stimulation of mPFC-to-AM depends on DA. TH-Cre mice received AAV-ChR2 into the mPFC, a probe implantation at the AM, and AAV-DIO-hM4D(Gi) into the VTA (Fig. 7e). After being trained to respond for photostimulation delivered into the AM over 5–6 sessions, effects of the DREADD agonist JHU 37160 (1 mg/kg, IP) and the $D_1$ receptor antagonist SCH 39166 (0.05 mg/kg, SC) were examined. Both JHU 37160 (1 mg/kg, IP) and SCH 39166 decreased ICSS rates (Fig. 7e, f). The stimulation sites in the AM and DREADD expressions in the VTA were verified (Supplementary Fig. 10c).

Given that the stimulation of the AM-to-mPFC pathway activated VTA DA neuron activity, we examined whether ICSS reinforced by this stimulation depends on VTA DA neurons. We used the mice trained to respond for the stimulation of the AM-to-mPFC pathway in the experiment shown in Fig. 6c. These mice received intra-VTA AAV-DIO-hM4D(Gi) when they received the surgery for intra-mPFC AAV-ChR2 and intra-AM probe (Fig. 7g). The inhibition of VTA DA neurons induced by JHU 37160 (1 mg/kg, IP) significantly decreased ICSS rates compared to vehicle (Fig. 7g), and the blockade of the $D_1$ receptor antagonist SCH 39166 (0.05 mg/kg, SC) also significantly decreased ICSS rates than the vehicle in the same mice (Fig. 7h). The stimulation sites of these mice are shown in (Fig. 6c and Supplementary Fig. 8c), and DREADD expressions in the VTA were verified and similar to those shown in Supplementary Fig. 10c.

**Reinforcing video clips activate mPFC and AM at the same time in humans.** We next examined how our findings might relate to human motivation. Similar to ICSS in which animals engage in self-initiated responding for stimuli that are not necessary for maintaining biological homeostasis, humans engage

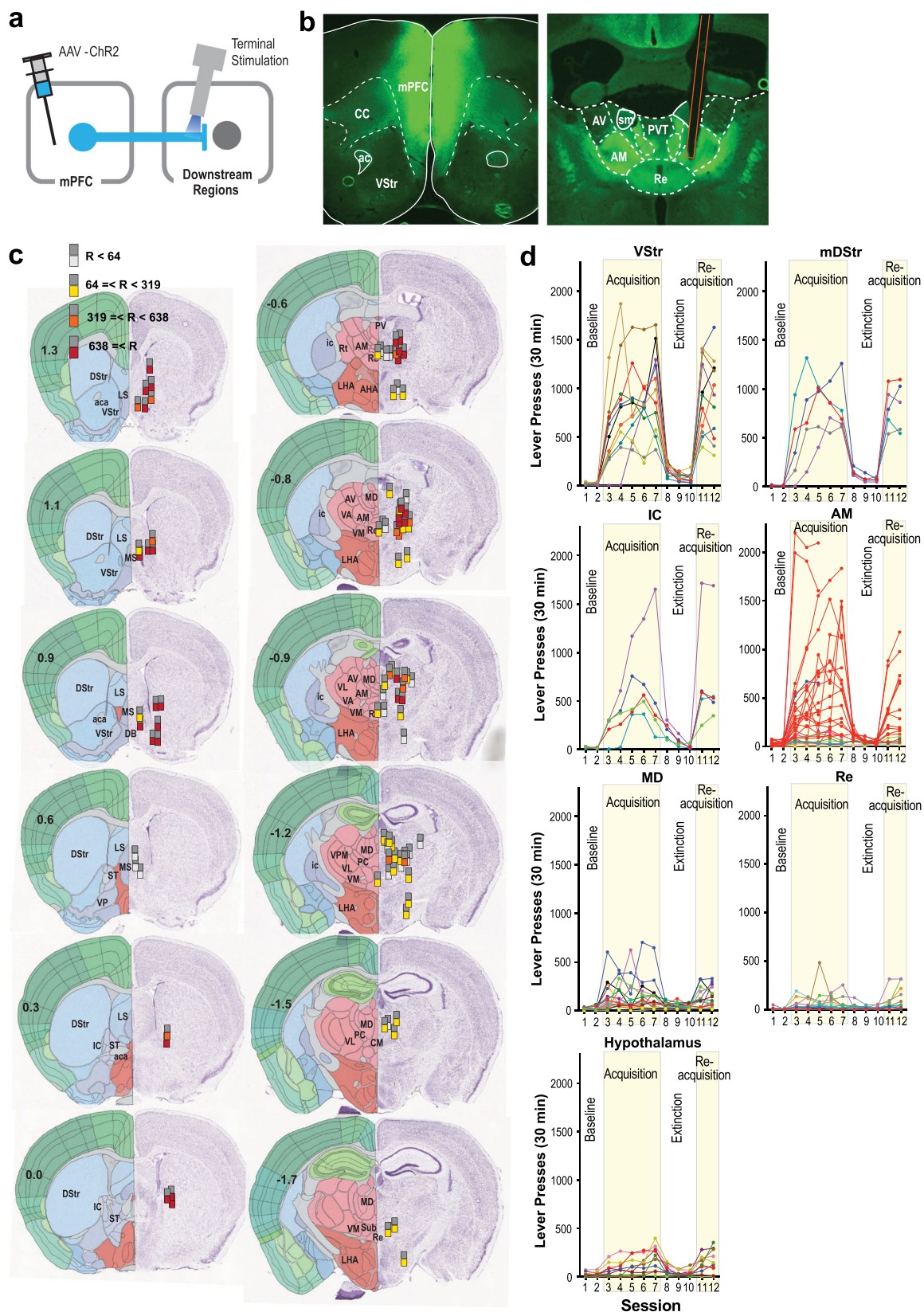

in self-initiated, binge responding with stimuli such as substances of abuse and video-watching. Recently developed computer algorithms such as that of the TikTok app can recognize video content, build a model of what each user likes to watch, and show the user more clips according to the model. TikTok-suggested video clips are known to be reinforcing and to induce binge viewing of successive clips[53]. Using fMRI, we examined whether TikTok-clip watching activates the mPFC and AM at the same time in humans. Each human participant watched TikTok video clips suggested by the TikTok algorithm model developed for the participant as well as TikTok clips assembled randomly without any model.

**Fig. 4 Axonal terminal stimulation of mPFC neurons supports ICSS in many downstream regions, including the anteromedial thalamic nucleus. a** Diagram showing the site of the injection of AAV-hSyn-ChR2-EYFP and regions of optic-fiber implantations. **b** Photomicrograms showing example expression of AAV-ChR2 in the mPFC (top) and in the anterior thalamus (bottom). Orange lines indicate the placement of optic fiber. **c** The sites ($n = 80$) and effectiveness of axonal terminal stimulation of mPFC neurons. Each dark-gray square indicates the tip of optic fiber, which is accompanied with a colored square below, indicating one of four levels of ICSS rates. Coronal sections were adopted from Allen Mouse Brain Atlas© of Allen Institute for Brain Science available at https://mouse.brain-map.org/static/atlas. aca anterior commissure, AHA anterior hypothalamic area, AM anteromedial thalamic nucleus, AV anteroventral thalamic nucleus, DB diagonal band of Broca, DStr dorsal striatum, IAM interanteromedial thalamic nucleus, ic internal capsule, LHA lateral hypothalamic area, LS lateral septum, MD mediodorsal thalamic nucleus, MS medial septum, NAc nucleus accumbens, PC paracentral thalamic nucleus, PV paraventricular thalamic nucleus, Re nucleus of reunion, Rt reticular nucleus, ST nucleus of stria terminalis, Sub submedius thalamic nucleus, VA ventral anterior thalamic nucleus, VL ventrolateral thalamic nucleus, VM ventromedial thalamic nucleus, VP ventral pallidum, VPM ventral posteromedial thalamic nucleus, VStr ventral striatum. **d** ICSS data of notable regions. An active lever press was rewarded with photostimulation (a 25-Hz 8-pulse train) in sessions 3–7 and 11–12. Each line shows lever-press data from a single mouse. The data of the AM stimulation site are shown in red, while the data of stimulation sites just adjacent to the AM are shown in a different color. Some mice with AM stimulation were only tested for the acquisition phase.

App-suggested clips activated the AM and BA9 of the mPFC in addition to various other brain regions including both ventral and dorsal visual streams, the frontoparietal network, bilateral anterior insula, posterior cingulate cortex, parts of middle brain and cerebellum, while they decreased BOLD signals in a set of brain regions including dorsal anterior cingulate cortex, middle cingulate cortex, and cuneus/precuneus, in contrast to rest (Fig. 8a). Control clips also activated similar regions, except the BA9 of the mPFC, which was less activated (Fig. 8b). The app-suggested clips induced higher activation in regions including the mPFC, AM, temporal gyri and posterior cingulate/precuneus area than the control clips (corrected $P < 0.05$) (Fig. 8c). Further region-of-interest (ROI)-based analyses suggest a stronger relationship between the mPFC and AM relative to mPFC and MD while viewing personalized video contents (Supplementary Fig. 11). The resting-state functional connectivity analysis with AM as seed (Supplementary Fig. 11a) shows connections with the entire cingulate cortex area and a large portion of the mPFC (corrected $P < 0.05$) (Fig. 8d). To determine the ROI within the mPFC, we chose the overlapping mPFC sub-area between the zone activated more by the app-suggested clips than the controls (Fig. 8c) and the AM-connected zone found with the resting-state analysis (Fig. 8d), resulting in BA9 and BA32 (Fig. 8e). Resting-state functional connectivity analysis using the mPFC ROI as a seed revealed a significant connection with the AM, in addition to the VStr and VTA, well-known reinforcement/motivation regions, and the posterior cingulate cortex and bilateral temporal gyri, which had displayed higher activation with video-clips watching (corrected $P < 0.05$) (Fig. 8f). Further ROI-based psychophysiological interaction (PPI) analysis[54] showed significantly higher BOLD signal coupling between mPFC and AM upon viewing app-suggested clips, compared to viewing control ones (PPI effect: mean (SEM) = 0.092 (0.029), $P < 0.005$). In addition, a dynamic causal modeling (DCM) analysis with parametrical Bayesian model reduction frame[55,56] indicates that the most reliable model ($P(m | Y) = 0.70$) shows reciprocal excitatory connections between mPFC and AM (Fig. 8g, h), rather than inhibitory or unidirectional excitatory connections. Moreover, the video-viewing task modulates the mPFC–AM system primarily through the disinhibition of mPFC ($P(m | Y) = 0.67$, Fig. 8i). These results support the idea that both the AM and the human mPFC (BA9 and BA32) are reciprocally connected in a positive-feedback loop and are coactivated at the same time during a motivated state.

## Discussion

This study provides strong support for the idea that mPFC regulates motivation for goal-directed behavior; we demonstrated that the mPFC is involved in goal-directed motivation more effectively than adjacent cortical areas[16,27,28] by providing systematic-comparison data. Several important concepts have emerged from the present study. First, the mPFC regulates goal-directed motivation via multiple pathways that include extra-basal ganglia regions. Second, the mPFC and the AM interact with each other in a positive-feedback manner for goal-directed motivation. Third, the pathways between the mPFC and the AM can regulate the activity of DA neurons, which in turn regulate their motivation effects. In addition, we found that some mPFC neurons have collateral projections between AM and VTA and between AM and VStr; therefore, mPFC neurons appear to regulate activities between the basal ganglia and extra-basal ganglia regions for goal-directed motivation. These structures most likely form a network for goal-directed motivation because the mPFC receives feedback from the AM and the basal ganglia. Before discussing the emerging concepts and associated issues, we should discuss behavioral observations on ICSS involving mPFC stimulation, showing how mPFC stimulation is motivating.

Our experiments suggest that mPFC stimulation is highly motivating or reinforcing. Experimentally naive mice quickly learned to increase responding on a lever that was contingently reinforced by mPFC stimulation. Optogenetic mPFC ICSS rates were essentially as high as those of ICSS with VTA DA neurons that we found in our previous studies using a comparable procedure[57,58]. This may mean that mPFC stimulation is as motivating or reinforcing as direct stimulation of DA neurons, which are considered a key neuronal population for goal-directed motivation[20]. Consistently, when the reinforcement contingency was switched between the two levers, mice quickly abandoned a lever response that no longer provided mPFC stimulation and switched to a new lever that provided the reinforcement. It is also important to note that, despite vigorous engagement in ICSS with a continuous-reinforcement schedule, mice were not able to maintain the rate of responding when they had to make even one extra response for each reinforcement (i.e., mPFC stimulation train) or when they had to wait for more than 1 s for the next reinforcement, suggesting the transient nature of mPFC stimulation on motivating effect (Supplementary Fig. 2). Such effect of mPFC stimulation is quite a contrast to that of natural rewards such as food for which mice respond persistently and readily increase responses[57]. The fact that the pulse frequency of the train was also found to be critical in ICSS rates underscores the temporary nature of the stimulation on motivating effect. Such a transient effect of mPFC stimulation on ICSS was parallel to the transient effect of BOLD signals induced by mPFC stimulation. Similar transient effects of the stimulation of VTA DA neurons on ICSS have been previously reported[57].

Our experiments demonstrate that mPFC projections to extra-basal ganglia regions mediate motivation and reinforcement. It is

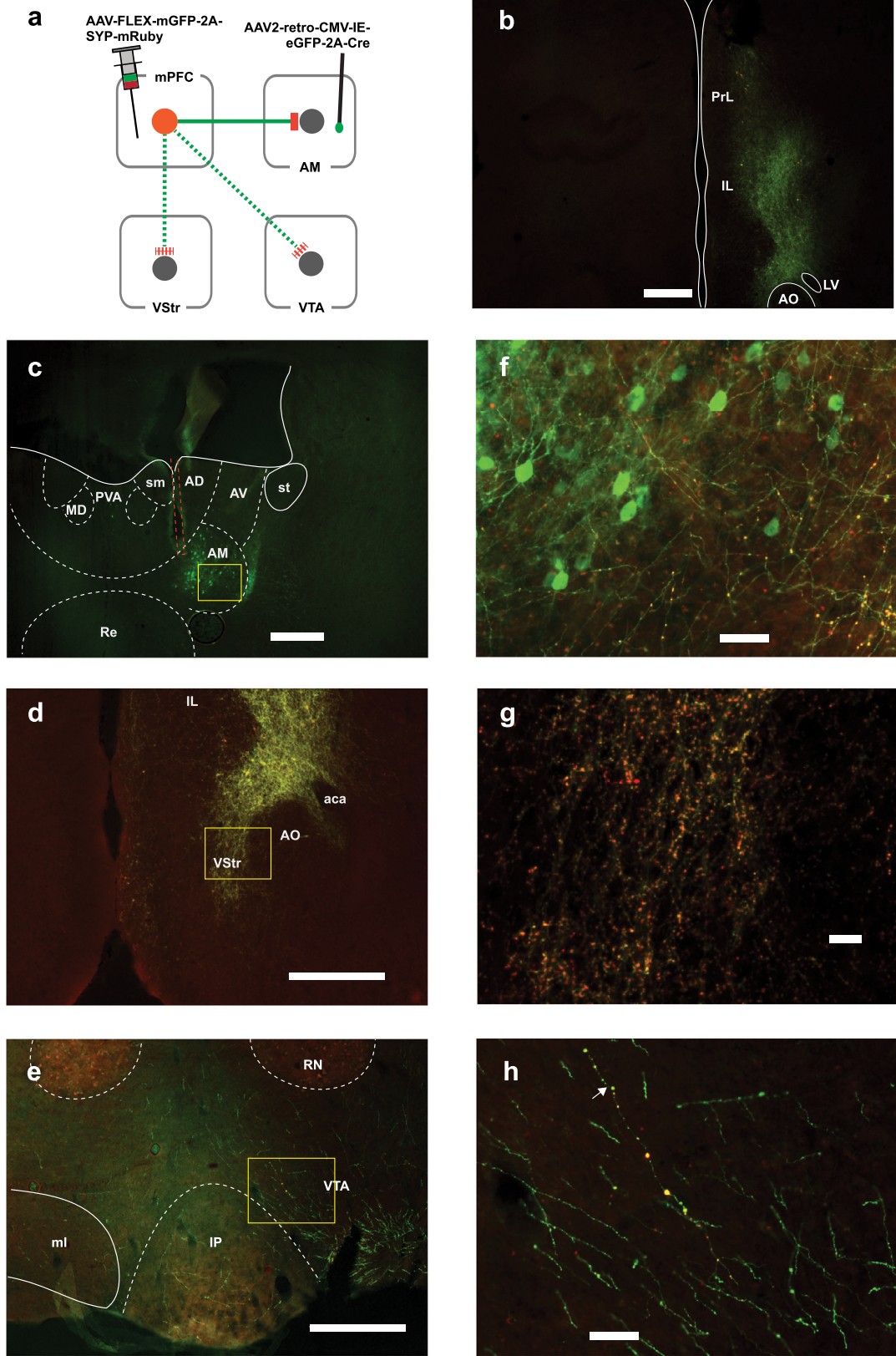

established that mPFC projections to the basal ganglia are important in goal-directed motivation (Fig. 9a): Response-contingent stimulation of the mPFC-to-VStr pathway[19] or PFC-to-VTA pathway[17] triggers ICSS, among other findings. Our rodent fMRI procedure revealed that mPFC stimulation increased activation in the VStr. However, the procedure did not detect the activation of VTA, even though robust VTA activation was detected by fiber photometry following mPFC stimulation. We entertain several reasons: first, the MRI receive coil was optimized for the detection of forebrain signals, but not those of inferior brainstem regions including the VTA; second, mPFC stimulation may have activated both VTA DA neurons and GABAergic

**Fig. 5 mPFC neurons projecting to the AM have collateral projections to the VStr and VTA (mouse 282). a** Diagram showing viral manipulations with intra-AM AAV-retro eGFP-Cre and intra-mPFC AAV-FLEX-mGFP-2A-SYP-mRuby. This experiment was repeated in three mice: The data of representative mice are shown here (**b–e**); the other two cases are shown in Supplementary Figs. 6 and 7. **b** Photomicrogram showing mPFC affected by the injections. **c–e** Photomicrograms showing the area of interest: AM (**c**), VStr (**d**), and VTA (**e**). The orange lines (**c**) indicate the cannula track. Scale bar = 250 μm. **f–h** Photomicrograms, enlarged areas inside of the rectangles (**c–e**), respectively, showing GFP-labeled fibers and mRuby-labeled boutons. Scale bar = 25 μm. The arrow (**h**) indicates a fiber that clearly contains mRuby-labeled boutons. aca anterior commissure, AD anterodorsal thalamic nucleus, AM anteromedial thalamic nucleus, AO anterior olfactory area, AV anteroventral thalamic nucleus, fr fasciculus retroflexus, IAM interanteromedial thalamic nucleus, IL infralimbic cortex, IP interpeduncular nucleus, LD lateral dorsal thalamic nucleus, LV lateral ventricle, MD mediodorsal thalamic nucleus, ml medial lemniscus, MHb medial habenula, MS medial septum, PrL prelimbic cortex, PV paraventricular thalamic nucleus, Re nucleus of reunion, RN red nucleus, sm stria medullaris, SNr substantia nigra, parts reticulata, VAL ventral anterior-lateral thalamic complex, VStr ventral striatum.

neurons that inhibits DA neurons, resulting in no BOLD signal; third, the rats were anesthetized; fourth, interactions of these conditions. Our rodent fMRI procedure also revealed various other downstream regions, including the septum, the thalamus and the hypothalamus. Our optogenetic investigations found that mice engaged in ICSS with the mPFC's terminals not only in the VStr and dorsal striatum but also in the internal capsule, which carries cortical signals to subcortical structures, the thalamus, and the hypothalamus. The hypothalamus or the mediodorsal thalamic region supported relatively low rates of ICSS, but septal stimulation did not support ICSS. However, we found that the stimulation of the anterior thalamic area, particularly the AM, supported relatively high rates of ICSS.

This study has provided substantial evidence that the AM is one of the structures through which the mPFC interacts in goal-directed motivation (Fig. 9a). Previous studies implicated the ATh, which includes the AM, in attention[59,60]. We believe that attention as a descriptive concept of AM functions is consistent with its motivational role because goal-directed motivated behavior is not tenable without attention (i.e., an essential component of goal-directed motivation). The present study took the advantage of optogenetics, which can selectively manipulate pathways and thereby enabled us to distinguish the AM from other ATh nuclei because the AM, but not the other ATh nuclei, sends efferents to and receive afferents from the mPFC[43,45,61]. We found that response-contingent stimulation of mPFC-to-AM, AM, and AM-to-mPFC neurons triggered ICSS. This positive-feedback arrangement of the mPFC–AM interaction is supported by our slice electrophysiology data and our human fMRI data.

We observed differential activations within the human mPFC as well as interactions among the mPFC, AM, VStr, and VTA during the video-viewing. Concerning this observation, we discuss three notable issues. First, discussed is anatomical specificity issue. Although we are confident that the AM is involved in motivational effects in mice because of consistent findings among multiple experiments targeting at the AM, human AM was examined by a single experiment. Like rodent AM, human AM is surrounded by other distinct thalamic nuclei. While increased BOLD signals are clearly detected in the AM, increased signals are also detected in surrounding nuclei. Therefore, the role of human AM in motivated behavior should be further substantiated by future studies. Nevertheless, such AM activation during motivated behavior is consistent with the observations in rodent experiments (further discussed below).

The second issue concerns functional and structural heterogeneity of the mPFC, whose sub-areas are differentially activated depending on the stimulus that participants perceive, the action that participants engage, or both. First, the video clips that the participants watched largely consisted of social stimuli, which may be processed in certain regions of the mPFC[62]. Second, the participants were simply asked to watch videos, but not physically interact with them. Thus, such conditions of the task may have differentially affected mPFC sub-areas as well as downstream

regions. In addition, it is important to consider the mPFC layers for understanding how they interact with downstream regions. mPFC neurons have multiple cellular layers, which are differentially connected with the downstream regions (Fig. 9b). The AM receives the largest mPFC afferents from layer 6; the VStr receives mPFC afferents from multiple layers; and the VTA exclusively receives mPFC afferents from layer 5[63]. The fMRI task may have selectively activated mPFC neurons in layer 6, which is reciprocally linked with the AM, but has little or no efferents to the VStr or VTA, an idea that could explain the observation that the video-watching task activated the mPFC and AM, but not the VStr or the VTA. Although such topic is beyond the scope of the present study, which focused on large-scale circuits, it is important for future research to consider how such microcircuit organization of the mPFC as well as how the sub-areas of the mPFC regulate motivated behavior in differential circumstances.

Finally, the human fMRI data also raise the issue of species differences between humans and rodents. The stimulation of rat IL activated the entire mPFC, and the stimulation of mouse mPFC sub-regions did not make a detectable difference in ICSS. We believe that it is important to distinguish between stimulation-induced stereotyped behavior and environmental event-induced adaptive behavior. The rate of ICSS is so stereotyped that it may not reflect the repertoire of behavior that mPFC sub-regions regulate and that this measure may simply capture a common functional property of mPFC sub-regions. Moreover, as discussed above, enlarged human mPFC may reflect an extended repertoire of behavior that mPFC sub-regions regulate.

VTA DA neurons are known as a key structure involved in goal-directed motivation[20,64,65]. Our fiber-photometry data suggest that the mPFC–AM loop can regulate DA neurons. We found that the stimulation of not only the mPFC-to-AM pathway, but also the AM-to-mPFC pathway, activated VTA DA neurons. Taken together, mPFC neurons projecting to the AM may activate AM-to-mPFC neurons, and in turn they may activate not only mPFC-to-AM neurons, but also mPFC-to-VTA neurons and even mPFC-to-VStr neurons for the activation of VTA DA neurons (Fig. 9). In addition, the inhibition of VTA DA neurons decreased ICSS rates. We propose that the mPFC–AM interaction plays an important role in VTA DA neurons and participates in the global brain network for goal-directed motivation.

Our findings on the AM's role in motivation corroborate several clinical observations showing that patients with infarcts in or in the vicinity of the AM displayed apathy (also known as abulia), a condition with the quantitative reduction in goal-directed behavior[66–68]. Consistently, other clinical studies suggest that the dysregulation of the mPFC leads to apathy[69]. These clinical observations together with the present observations raise the possibility that the mPFC–AM interaction is a key component of the brain network regulating normal goal-directed motivation. It is of interest and potential importance to further investigate how the mPFC–AM loop participates in motivated behaviors and whether the mPFC–AM loop participates in motivational disorders such as

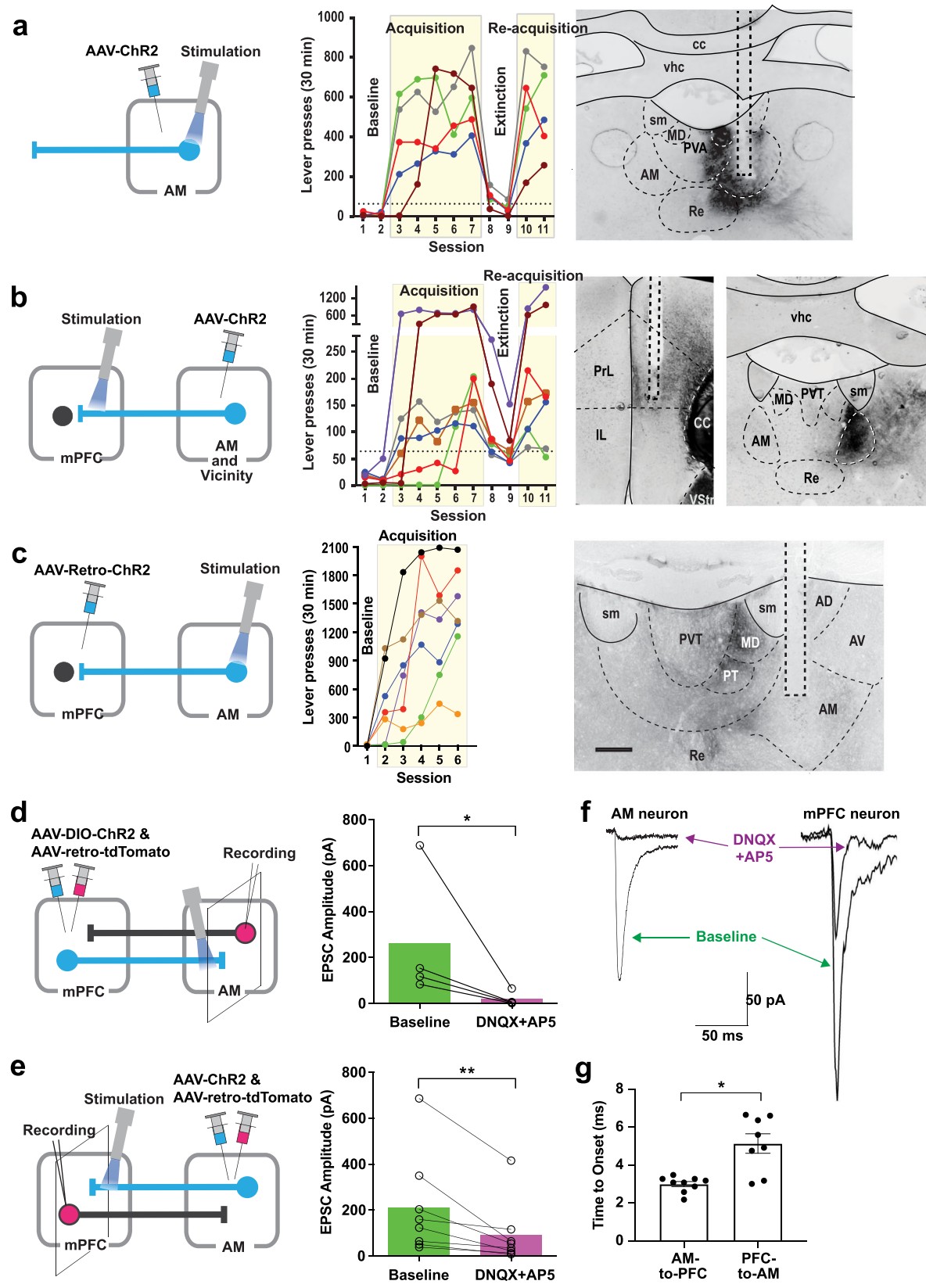

addiction and depression, which have begun to be treated with PFC stimulation procedures such as DBS and TMS.

## Methods

**Animals**. Male and female C57BL/6J mice were purchased from Jackson Labs (Bar Harbor, ME). Transgenic mice (VgluT1-Cre, VgluT2-Cre, Vgat-Cre, and Emx1-Cre) were purchased from Jackson Labs, and these mice and TH-Cre mice[70] were crossed with C57BL/6J and bred at the NIDA Transgenic Breeding Facility. Wister Han outbred rats were purchased from Envigo (Indianapolis, IN). The mice and rats were individually housed in a colony maintained at a consistent temperature (70–74 °F) and humidity (35–55%) with a 12 h light/dark cycle (lights on at 7 am) and had ad libitum access to food and water except during testing. This study used male and female mice and male rats with 2.5–6 months of age at the time of testing.

**Fig. 6 AM neurons support ICSS and form a positive-feedback loop with mPFC neurons. a–c** Left: diagrams showing the sites of the AAV-ChR2-EYFP injections and optic-fiber implantations. Middle: ICSS data: Responding on the active lever was rewarded with photostimulation (a 25-Hz 8-pulse train) in sessions 3–7 and 10–11, while no photostimulation in sessions 1–2 or 8–9. Each line shows lever-press data from a single mouse. Right: photomicrograms showing example EYFP expressions and optic-fiber tracks. AM projections were found throughout the mPFC with sparse expression. Stronger anterograde expression was tended to be found in the PrL than the IL and the anterior PrL than posterior PrL, and the two highest responders had their placements in the anterior PrL (**b**). Scale bar = 250 μm. **d, e** Left: diagrams showing slice electrophysiology preparations with injection sites and stimulation sites. Emx1-Cre and wild-type C57Bl6 mice ($n = 7$) were used for the experiments described in (**d**) and (**e**), respectively. Right: stimulation-evoked EPSC. *$P = 0.0017$ ($t_3 = 10.83$) and **$P = 0.0010$ ($t_7 = 5.38$), ratio paired, two-tailed $t$-test. **f** Example traces: On the left, an AM neuron showing a light-evoked response when adjacent PFC terminals are activated. The response is diminished with the application of glutamatergic antagonists. On the right, a PFC neuron showing a light-evoked response when adjacent AM terminals are activated. The response is diminished with application of glutamatergic antagonists. **g** Mean (±SEM) and individual latencies to response onset as defined as time to achieve 10% of the peak response. *$P = 0.0007$ ($t_{15} = 4.22$; $n = 9$ AM cells; $n = 8$ mPFC cells), unpaired, two-tailed $t$-test.

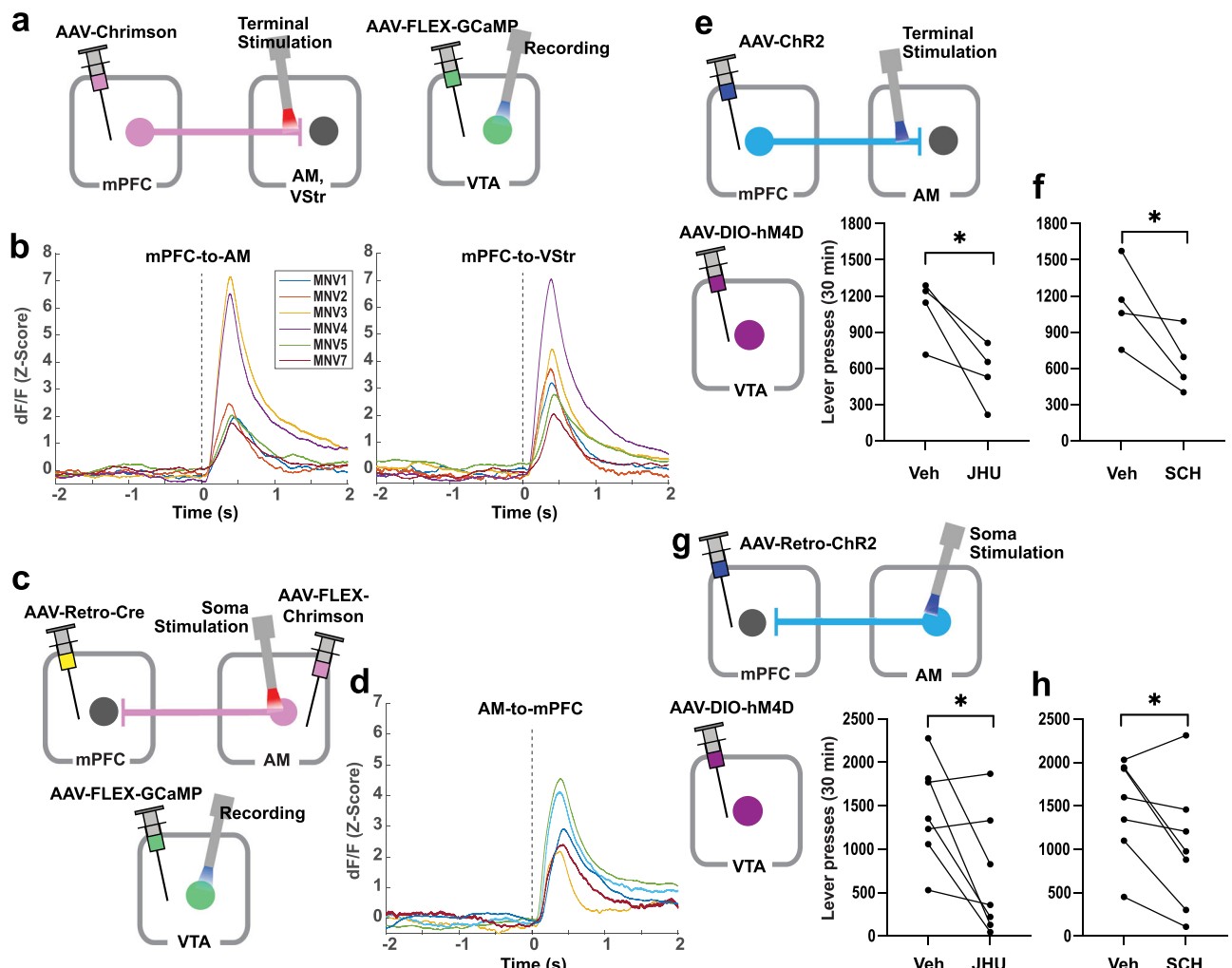

**Fig. 7 mPFC-to-AM neurons or AM-to-mPFC neurons regulate the activity of VTA DA neurons. a** Schematic diagram showing AAV-Chrimson injection into the mPFC and stimulation at the AM or VStr (left) and AAV-FLEX-GCaMP injection and recording at the VTA (right) in TH-Cre mice. **b** GCaMP signals ($Z$-score dF/F) measured in the VTA as a function of 8-pulse photostimulation of the mPFC-to-AM (left) and mPFC-to-VStr (right) pathways in freely moving mice. Colored lines indicate data of six mice. The dotted line at 0 s indicates the onset of the photostimulation train. **c** Schematic diagram showing injections of AAV-FLEX-Chrimson, AAV-retro-Cre, and AAV-FLEX-GCaMP into the AM, mPFC, and VTA, respectively, in TH-Cre mice. Stimulation and recording probes were inserted into the AM and VTA, respectively. **d** GCaMP signals ($Z$-score dF/F) measured in the VTA as a function of 8-pulse photostimulation of the AM-to-mPFC pathway in freely moving mice. Colored lines indicate data of five mice. The dotted line at 0 s indicates the onset of photostimulation train. **e** Schematic diagram showing AAV-ChR2 injection into the mPFC, stimulation probe at the AM, and AAV-DIO-hM4D injection into the VTA in TH-Cre mice. *$P = 0.034$ ($t_3 = 2.79$, $n = 4$), paired, one-tailed $t$-test. **f** *$P = 0.035$ ($t_3 = 2.74$, $n = 4$), paired, one-tailed $t$-test. **g** Schematic diagram showing AAV-Retro-ChR2 injection into the mPFC, stimulation probe at the AM, and AAV-DIO-hM4D injection into the VTA in TH-Cre mice. *$P = 0.023$ ($t_6 = 2.50$; $n = 7$), paired, one-tailed $t$-test. **h** *$P = 0.016$ ($t_6 = 2.78$, $n = 7$), paired, one-tailed $t$-test.

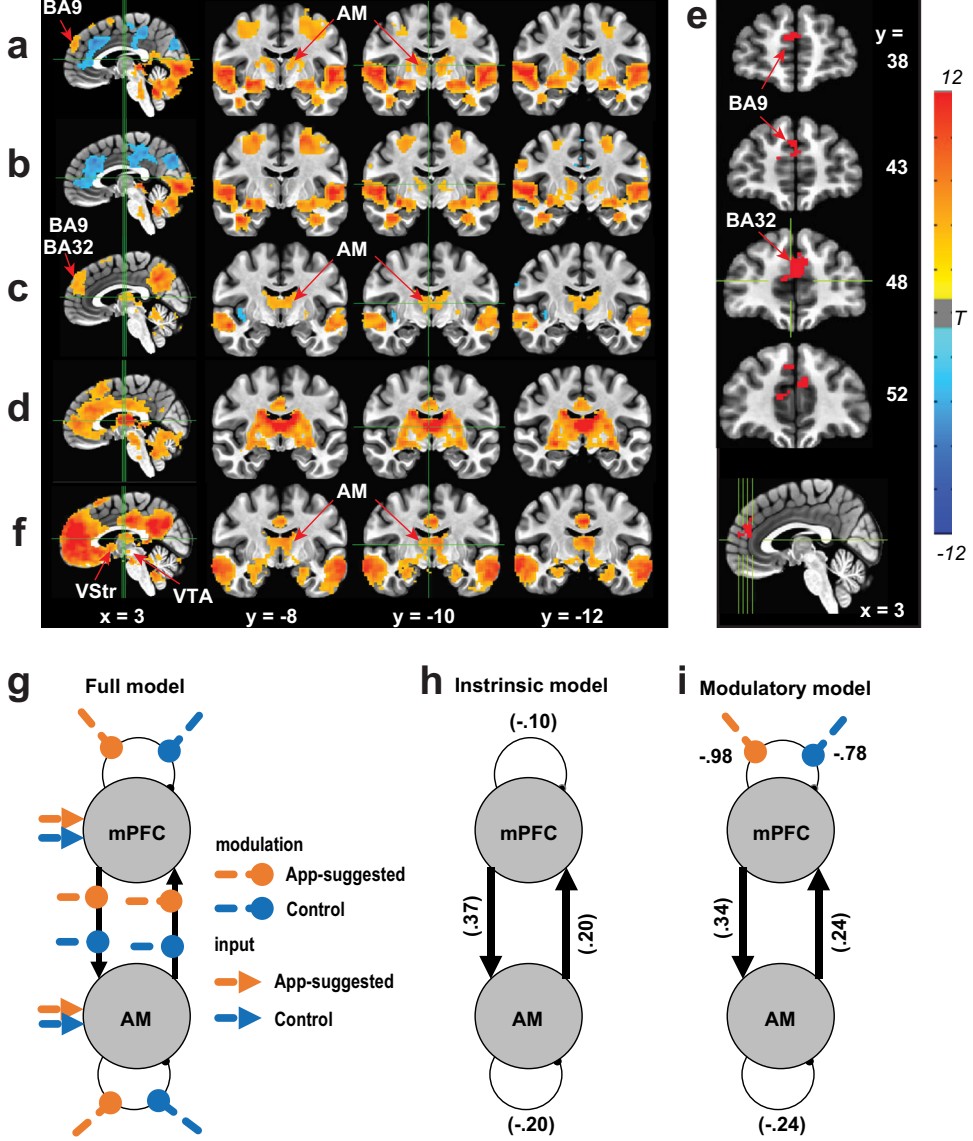

**Fig. 8 Reinforcing video clips activate mPFC and ATh at the same time in humans. a** Change in BOLD signals upon viewing App-suggested video clips. **b** Change in BOLD signals upon viewing control clips. **c** Difference in BOLD signals between app-suggested clips and control clips. **d** Resting-state functional connectivity map with AM as seed. **e** Overlapping mPFC zone (BA32 and 9) between **c** and **d**. **f** Resting-state functional connectivity map of the mPFC (**e**) as seed. **g** Dynamic causal modeling analysis on the mPFC–AM circuit with a full-model structure involving three components consisting of four intrinsic connections, eight experimental modulation on these connections, and four direct inputs from experimental conditions. **h** Empirical Bayesian model reduction on the four intrinsic connection parameters indicated the fully connected model was the best intrinsic model (P(m|Y) = 0.70) in which mPFC and AM showed excitatory between-region connections and inhibitory within-region connections. **i** Empirical Bayesian model reduction on the sixteen full-model parameters showed the best model (P(m|Y) = 0.67) in which parameters with posterior probability >0.95 include: (1) intrinsic positive input from mPFC to AM; (2) intrinsic positive input from AM to mPFC; (3) self-inhibition of AM; (4) disinhibition of mPFC by App; and (5) disinhibition of mPFC by control-video viewing.

All procedures were approved by the Animal Care and Use Committee of the Intramural Research Program of the National Institute of Drug Abuse and were in accordance with the National Research Council Guide for the Care and Use of Laboratory Animals. To acclimate animals to experimenters, experimenters handled animals for 5–10 min a day for 3–4 days with or without fiber cables connected to them prior to the start of behavioral testing.

**Viral vectors.** AAVs, which are described in Supplementary Table 1, were obtained from the NIDA Genetic Engineering and Viral Vector Core (GEVVC; Baltimore, MD), Addgene (Watertown, MA), and UNC Vector Core (Chapel Hill, NC).

**Intracranial surgeries**. Mice were anesthetized with either ketamine/zylazine mixture (80/12 mg kg$^{-1}$, i.p.) or isoflurane (1–2%) for stereotaxic surgeries. Each mouse typically received unilateral injection of a viral vector (200–500 nl) and an

optic-fiber implantation unless noted otherwise. The coordinates for viral vector injections are summarized in Supplementary Table 2. The optical fiber was implanted 0.2 mm above the injection site.

**ICSS general procedure**. Experimentally naive C57BL/6J mice and rats were individually placed in operant conditioning chambers equipped with two retractable levers, cue lights above the levers, and a house light. They were habituated to the chamber for 30 min with levers retracted, followed by one or two baseline sessions in which no photostimulation was delivered upon lever pressing. Following these sessions, each animal was simply placed in the chamber where responding on the "active" lever resulted in intracranial delivery of a 8–16-pulse light train (473-nm blue-light, 3-ms pulse-duration delivered at 25–50 Hz, and ~7 mW laser power at the tip of stimulation sites), while responding on the "inactive" lever had no programed consequence. A 25-Hz 8-pulse train was used for most experiments to excite PFC neurons because PFC principal neurons can

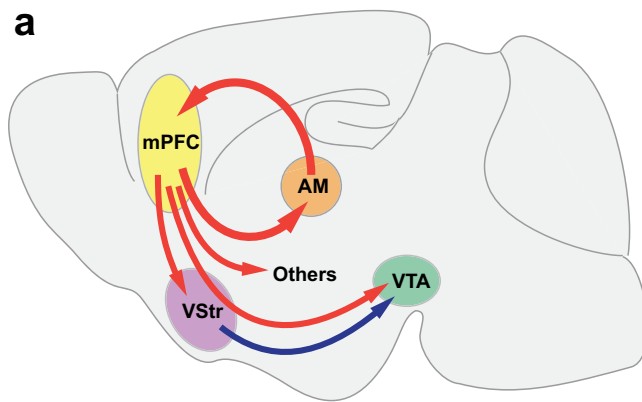

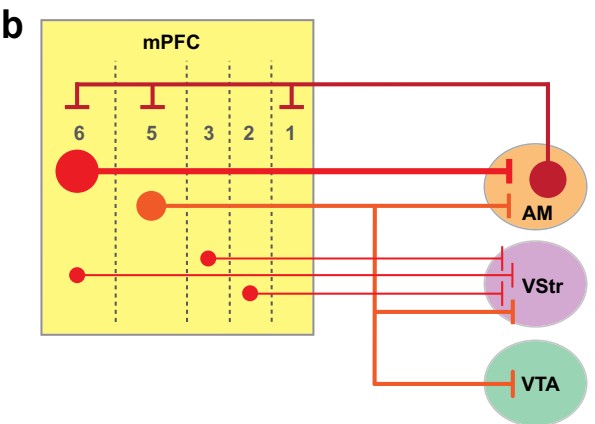

**Fig. 9 The mPFC regulates goal-directed motivation and dopamine activity via multiple projection pathways. a** Schematic drawing of the sagittal brain section showing the mPFC–AM loop and other key mPFC pathways to regions that participate in regulating goal-directed motivation and DA activity. **b** Schematic drawing showing a microcircuit model between the mPFC and its downstream projection regions of the AM, VStr, and VTA.

display spiking activity at 20 Hz or slightly over[71]. The assignment of active and inactive levers between the two levers typically stayed constant throughout experiments unless stated otherwise. Each session lasted for 30 min, and sessions are typically separated by 1 d.

We used the lever-pressing rates of the control mice to establish non-reinforced response levels. The control mice ($n = 8$) had the mean of 21.8 presses with the standard deviation (SD) of 13.8 per session during sessions 3–7 (Fig. 1d), which led us to derive 64 (mean + 3 * SD) presses or greater in a 30-min session as ICSS responders.

**ICSS: reversal of active and inactive lever assignment test.** The mice with mPFC implants ($n = 9$) used in the ICSS experiment described above received a lever reversal test over three sessions in which the assignment of active and inactive levers with respect to the right and left levers was reversed. The lever assignment of the first session was consistent with previous sessions, and the lever assignment was reversed in the 2nd session without any cue. The assignment in the 3rd session was the same as that of the 2nd session.

**ICSS: frequency test.** We examined the effects of stimulation frequency at 6.25, 12.5, 25, and 50 Hz in counterbalance order over a 40-min session with the high responders (>1000 lever presses in 30 min) that were used for the reversal test above ($n = 9$). Each frequency's effects were examined in a 10-min period (or two bins of 5 min); the number of presses during the second bin was used to represent the effect of the specific frequency. We also calculated how fast each mouse would have responded if they had responded for 6-, 13-, and 25-Hz trains with the same interval that they responded for 50-Hz trains: 300-s bin/ (response interval for 50-Hz trains + train length of 0.1172, 0.577 and 0.283 s for the 6-, 13- and 25-Hz frequency, respectively).

**ICSS: lever ratio test.** Following the frequency test, the effects of response ratio schedules of reinforcement were examined. The mice ($n = 9$) received mPFC stimulation upon either 1, 2, or 4 lever presses. The lever-press requirement was changed in an ascending order every 10 min. This procedure was repeated for three sessions.

**ICSS: interval test in rats.** The effects of stimulation interval (i.e., duration that rats ($n = 7$) had to wait before responding for stimulation) at 0, 1, 2, and 4 s in counterbalance order over a 40-min session. Each interval's effects were examined in a 10-min period.

**Electrophysiological recording of MPFC neurons.** An optic fiber and four or eight tetrodes were combined to make an optetrode as previously described[72]. C57BL/6J Mice ($n = 3$) received an injection of AAV-ChR2 into the IL and a movable optetrode just above the IL. Trains of eight pulses at 6.25, 12.5, 25, and 50 Hz were delivered at a random order on a variable-interval schedule with a mean of 8.5 s ranging between 7 and 10 s.

**Real-time place-preference test with unilateral photomanipulation.** The same test chamber as described in the above ICSS experiments with levers retracted and was divided into two equal-size compartments by placing a Plexiglas barrier with the height of 12 mm from the grid floor. To further help mice to distinguish between the compartments, a 5 kHz tone was continuously delivered when mice were in the right compartment, while a 10 kHz tone was delivered when mice were in the left. Wild-type mice with intra-mPFC AAV-NpHR ($n = 6$) and their control counterparts ($n = 6$) received continuous photostimulation via the implanted optic fiber while in one of, but not the other, compartments. These mice received photostimulation in the right compartment for the first five sessions, no photostimulation in session 6, and photostimulation in the left for the last five sessions. The assignment of tones to the compartments was kept constant throughout the experiment. Each session lasted for 30 min, and sessions were typically separated by 1 d.

**Effects of the bilateral inhibition of mPFC neurons.** Mice (NpHR: $n = 5$; EYFP: $n = 4$) were run through a series of behavioral tests to examine the effects of bilateral inhibition of mPFC on various behaviors. For all tests, 3.5–5 mW of green (532 nm) light was continuously delivered through the implanted optic fiber for the specified durations. Activity and animal positions were determined automatically using the video tracking software EthoVision XT v15 (Noldus).

*Real-time place-preference test with bilateral photomanipulation.* The mice were attached to optical lines and placed in a custom-made chamber with a plastic divider separating the chamber into two halves. Visual cues (orange horizontal or blue vertical bars) adorned the sides of the two halves and the orientation of this was counterbalanced. Fresh bedding was used for each session. For the first session (baseline), no light was delivered and for the next three sessions light was delivered when the animal was in the "left" half of the chamber only. Animal position within each side of the chamber was tracked and recorded with EthoVision XT v15.

*Forced swim test.* A 4000-mL Pyrex beaker was filled with water (25 °C) to a depth of 6 inches. The mice were connected to optical lines and gently placed into the water. Green light was delivered for the duration of the 6-min trial. The video was recorded from the side and immobility behavior was automatically scored using video tracking software. Only the last 4 min of the session was analyzed[73].

*Open-field test.* The mice were attached to optical lines and placed in one corner of a square Plexiglass arena ($42 \times 42$ cm with $21 \times 21$ cm center) with bright overhead lights. Green light was delivered continuously for the duration of the 5-min session. The video was recorded and the time spent in the center vs the border of the arena was analyzed with EthoVision XT v15.

*Novel environment exposure for c-Fos counting.* To test the efficacy of the NpHR-mediated inhibition of cells in mPFC using our light parameters, we placed animals for 15 min in a novel environment, consisting of a round Plexiglass bowl-shaped container (40 mm in diameter) with clean wood chip bedding and a novel object (small ball-shaped plastic toy). The green light was turned on before animals were connected to the fiber optic lines and were continuous for the duration of the session. We then placed the mice back in their home cage and waited 45 min before sacrificing and harvesting brains for c-Fos counting.

*c-Fos Immunohistochemistry and cell counting.* After completion of the behavioral experiments, all mice were intracardially perfused with ice-cold 0.9% saline followed by 4% paraformaldehyde. Brains were coronally sectioned at 40 μm. Brain sections were processed for immunohistochemistry detecting c-Fos using rabbit anti-c-Fos (1:3000; no. AB152MI, Santa Cruz Biotechnology) and goat anti-rabbit AlexaFluor 594 (1:300; Life Technologies) primary and secondary antibodies, respectively. c-Fos expression was captured with a ×5 lens on a fluorescent-microscope/video/computer system and quantified $500 \times 500$ μm area placed in

**Table 1 Demographic information.**

| Subject number | $N = 25$: $M = 11$; $F = 14$ | | |
|---|---|---|---|
| Age: mean (std.)/range | 24.1 (2.48) | Between 21 and 30 years old | |
| Use mode (%) | 72: viewing only | 28: viewing & publishing | |
| Use history (%) | 36: less than 0.5 years | 32: 0.5–2 years | 32: more than 2 years |
| Time spent per day (%) | 56: less than 1 h | 36: 1–2 h | 8: 2–4 h |

mPFC at the level of the anterior horns of the corpus callosum. We counted cells in sections corresponding to the tip of the optic fiber and 1 to 2 sections anteriorly before the optic fiber for all mice. c-Fos-ir cells were detected automatically and counted with ImageJ v1.51 (NIH).

**fMRI in rats with mPFC stimulation**. Rats were anesthetized with isoflurane and received AAV1-hSyn-ChR2(H134R)-EYFP ($n = 10$) or AAV1-hSyn-EYFP ($n = 7$) injections (0.5 μl) into the infralimbic region of the MPFC and an optic fiber just above the injection site. Stereotaxic coordinates for injections are: AP, 3.2; ML, 0.6; DV, 5.2 mm from the skull surface, and optic fibers were placed 0.4 mm above. Three weeks later, the animals were placed in an operant conditioning chamber and trained to press a lever for optogenetic self-stimulation. A 25-Hz stimulation train consisted of eight pulses (3-ms pulse-duration) was delivered upon pressing the lever.

The rats then underwent fMRI scanning on a Bruker Biospin 9.4 T scanner using a protocol detailed in a previous study[74]. During the scanning, animals were kept anesthetized with a combination of isoflurane (0.5%) and dexmedetomidine hydrochloride (0.015 mg kg$^{-1}$ h$^{-1}$). Block-design optogenetic stimulation was delivered to the right mPFC under three conditions: 25-Hz trains with an interval of 1, 2, or 4 s, respectively (Fig. 3c). Each condition consisted of five blocks, and each block consisted of 20 s stimulus on and 40 s off. Two scan sessions with the stimulus order of (A) 1s-2s-4s-interval or (B) 4s-2s-1s-interval were performed. The order of scans A and B was counterbalanced between animals. BOLD fMRI data were acquired using a T2*-weighted EPI sequence (TE = 13 ms, TR = 1000 ms, segment = 2, FOV = 35 × 35 mm$^2$, matrix size = 64 × 64, slice thickness = 1 mm, slice number = 15). FMRI data were preprocessed with slice timing correction, head motion correction, spatial smoothing (full-width-at-half-maximum 1.25 mm), and normalization. Voxel-wise whole-brain activation was then analyzed using a general linear modeling (GLM) approach, in which the response vector was BOLD signal of each voxel, and predictors were the three boxcar functions (defining the three types of stimulation) convolving with canonical hemodynamic response function (HRF), together with six head motion parameters and low-frequency drifts (i.e., linear and quadratic changes with scanning time) as nuisance variables.

Independent two sample $t$ tests were conducted to examine brain areas showing differences in activation between the two groups. Results were corrected for multiple comparisons using randomization and permutation simulation in AFNI's program 3dttest++ to achieve corrected $P < 0.05$, which was determined by single voxel $P$ value < 0.005 and minimum cluster size = 20 voxels (Fig. 3d, Supplementary Fig. 4). The relationship between the fMRI activation and reward behavior was examined using a correlation analysis by correlating beta values of each stimulus type resulting from GLM with lever press measured outside the scanner when animals were awake.

**Human fMRI procedure**

*Subjects.* Twenty-five healthy students from Zhejiang University participated in this study (Table 1). All the participants have used the TikTok App and maintained an active account. Most participants (72%) only used the App for video-viewing, while the rest used the app for both viewing and publishing their own video clips. Sixty-four percent of the participants had used TikTok for more than half a year, and 44% reported that they spent more than one hour on watching videos with this app every day. All the participants had normal or corrected-to-normal vision and reported no neurological diseases. This study was approved by the local ethical committee of Zhejiang University. Written informed consent was obtained from every participant before the experiment.

*Procedure concerning video clips.* We used the video content-recommendation algorism of TikTok, a popular video app, to examine how individualized content presentation modulates brain activity. We obtained a separate signed consent form from each participant for privacy protection. We acquired individualized video clips (IVs; 6 min long) by signing in the account of each participant and recorded clips (Xiaomi, Model: MI9) right before the fMRI experiment. In addition, we acquired control-video clips (CVs; 6-min long), which were obtained through a newly registered account so that the TikTok app had no viewing history information or model for the suggestion. The same CVs, which included 29 clips ranging from 5 to 21 s were used for all the participants. We confirm that 91% of the participants preferred the IVs over CVs, although the rest preferred CVs over IVs.

*Experimental procedure during fMRI session.* The participants were instructed to be relaxed and watched video clips through an angled mirror and heard the sound-track through headphones. The presentation of these stimuli was controlled by the software E-prime 3.0 (https://pstnet.com/products/e-prime/). The experiment adopted a block design with IVs and CVs, each type of clip was presented in six 1-min blocks. The two types of the 6 blocks were presented alternatively and separated by a 30-s break, during which the participants viewed a white fixation on a black background. Half of the participants started with an IV block first, while the other started with a CV block first. To minimize disruption, no additional question or interruption was introduced during the experimental session.

*MRI data acquisition and preprocessing.* MRI data were collected using a Siemens 3.0-T scanner (MAGNETOM Prisma, Siemens Healthcare Erlangen, Germany) with a 20-channel coil. High-resolution anatomical images were acquired using a T1-weighted magnetization prepared rapid gradient echo sequence with parameters below: TR = 2300 ms, TE = 2.32 ms, voxel size = 0.90 × 0.90 × 0.90 mm$^3$, flip angle = 8°, field of view = 240 mm$^2$, voxel matrix = 256 × 256. FMRI data were collected using a T2*-weighted gradient echo-planar imaging sequence with multi-bands acceleration (TR = 1000 ms, TE = 34 ms, slice thickness = 2.50 mm, voxel size = 2.50 × 2.50 × 2.50 mm$^3$, voxel matrix = 92 × 92, flip angle = 50°, field of view = 230 mm$^2$, slices number = 52, MB-factor = 4). An 8-min resting-state fMRI data (480 scans) were collected after structural image acquisition, during which participants were instructed to watch a white fixation on a black screen with relaxation, not to think anything in particular, and keep still during the scan session. A total of 1095 scans for the task fMRI data were collected using the same acquisition parameters.

Preprocessing of fMRI data included the following steps. First, slice time correction and head motion correction were performed using AFNI[75] (https://afni.nimh.nih.gov/). Tissue segmentation was then conducted to extract brains using SPM12 (https://www.fil.ion.ucl.ac.uk/spm/). Structural and functional images were co-registered and normalized into the MNI space using ANTs (http://stnava.github.io/ANTs/). Finally, spatial smoothing was applied to the normalized fMRI data with a 5 mm full-width-at-half-maximum Gaussian kernel. For resting-state data, two more preprocessing steps were included: (1) nuisance variable regression including six-rigid head motion and their forward derivates, FD (see below), and the first 5 principal components from white matter and cerebral spinal fluid (CSF) separately; and (2) band-pass filtering (0.01–0.1 Hz) was applied.

*Head motion handling.* Following a previous study by Power et al.[76], framewise displacement (FD, Eq. (1)) of fMRI data was calculated for each participant as indices of head motion for the task and resting-state data separately.

$$FD_i = |\Delta d_{ix}| + |\Delta d_{iy}| + |\Delta d_{iz}| + |\Delta \alpha_i| + |\Delta \beta_i| + |\Delta \gamma_i| \tag{1}$$

where $\Delta d_{ix} = d_{(i-1)x} - d_{ix}$, and similarly for the other rigid body parameters [$d_{ix}$ $d_{iy}$ $d_{iz}$ $\alpha_i$ $\beta_i$ $\gamma_i$]. A participant would be excluded from statistical analysis if her/his mean FD > 0.3 mm, or the total number of frames with FD > 0.5 mm is more that 10% of the total length of the data. With these two criteria, two subjects were excluded from task fMRI data analysis, and one subject was excluded from resting-state fMRI statistical analysis. Head motions parameters were also included as nuisance variables in the first-level fMRI data analysis.

*First-level task fMRI statistical analysis.* For the first-level analysis, general linear modeling (GLM) was conducted using the command 3dDeconvolve within AFNI. The six blocks for IVs and six for CVs during encoding were convolved with hemodynamic function to create two regressors to assess activity elicited by the two conditions, respectively. Further, two event-related regressors were created to model the transient effect associated with the start and end of video blocks for IVs and CVs, respectively. The six head motion parameters were included in the model as covariates, together with eight polynomial variables to remove task-unrelated artifacts.

*Group-level task fMRI statistical analysis and results.* Voxel-wise one-sample $t$ tests on beta maps of the block regressors for IVs and CVs were conducted to assess brain activation related to the two conditions, respectively. Paired wise $t$-test was conducted to examine brain areas showing differences in brain activation under the two conditions. Results were corrected for multiple comparisons (corrected $P < 0.05$, Fig. 8a–c) using randomization and permutation simulation in AFNI (single voxel $P$ value < 0.001 and minimum cluster size = 33 voxels).

*Resting-state functional connectivity of AM and mPFC.* Although higher brain activation in the anterior thalamus was seen in the paired *t*-test, the main purpose of the present analysis was to investigate the implication of the same mPFC–AM circuit in human motivational behavior. Therefore, we defined AM (Supplementary Fig. 11a) by drawing ROI based on the "Atlas of Human Brain" 4th edition[77]. Five voxels in each hemisphere were identified with a spatial resolution of 2.5 mm³. Time course was extracted from the AM seed and correlated with the time course throughout the whole brain. Resultant correlation coefficients were transferred into *Z*-maps with Fisher's *Z*-transformation (Eq. (2)). The mPFC seed (Fig. 8e) was defined by the union of the AM-seeded resting-state connectivity map (Fig. 8d) and task activation region showing higher activity in the IVs condition (Fig. 8c). One-sample *t*-test was applied to the *Z*-maps and results were corrected for multiple comparisons (corrected $P < 0.05$, Fig. 8d, f).

$$z = 0.5[\ln(1 + r) - \ln(1 - r)] \tag{2}$$

*PPI analysis of mPFC–AM circuit.* Psychophysiological interactions (PPI) is based on statistical models of factorial design by substituting brain activity in mPFC regions for one of the factors[54]. Another factor is the categorical video type condition (IVs vs CVs) in this study. Equation (3) below illustrates the PPI model.

$$y = A * (C_1 - C_2) * \beta_1 + A * \beta_2 + (C_1 - C_2) * \beta_3 + X * \beta_4 + \epsilon \tag{3}$$

The effect of video-viewing condition is assessed with the contrast term $(C_1 - C_2)$, where $C_1$ and $C_2$ are coded as 1 to represent IVs and CVs viewing conditions, respectively. The term A is a physiological variable of mPFC (BOLD signal here), and similarly, y is the BOLD signal of AM region. While $A*(C_1 - C_2)$ indicates the PPI term, its regression coefficient $\beta_1$ is used to infer the significance of PPI effect. The term X denotes all nuisance variables and $\epsilon$ is the error term. In this study, nuisance variables include six head motion parameters and eight polynomial variables determined by 3dDeconvolve, and BOLD signals from the dACC region that showed deactivation during the task. Note that PPI and DCM analyses were conducted using SPM12. To keep consistency with results obtained with AFNI, the eight polynomial variables were included as nuisances and the cutoff for high-pass filtering was set to 1024 s instead of using the default 128 s when running GLM for PPI and DCM.

*DCM analysis of mPFC–AM circuit.* Dynamic causal modeling (DCM) is a technique to infer neural processes that underlie observed time courses[78], such as fMRI data. The DCM procedure involves the Bayesian estimation of parameters of neuronal system models from which BOLD signals are predicted through converting the modeled neural dynamics into hemodynamic response based on the neurovascular coupling model. Statistically, DCM models the temporal evolution of the neural state vector as a function of the current state (z), the task input (u), and parameters ($\theta^n$) that define the functional architecture and interactions among brain areas at the neuronal level (Eq. (4)).

$$\frac{dz}{dt} = F(z, u, \theta^n) = Az + \sum_{j=1}^{m} u_j B_j z + Cu \tag{4}$$

These parameters describe the nature of the three causal components which underlie the modeled neural dynamics: (1) the parameter set A in Eq. (5) describes intrinsic effective connectivity among brain regions (a $k \times k$ matrix), (2) the parameter $B_j$ (Eq. (6)) describes context-dependent changes in effective connectivity induced by the *j*th input $u_j$ (a $k \times k$ matrix for each input), and (3) the parameter C (Eq. (7)) describes direct task inputs into the system that drive regional activity (a $k \times m$ matrix), whereas k is the number of nodes in the system and m is the number of task inputs.

$$A = \frac{\partial F}{\partial z} = \frac{\partial \dot{z}}{\partial z} \tag{5}$$

$$B_j = \frac{\partial^2 F}{\partial z \partial u_j} = \frac{\partial}{\partial u_j} \frac{\partial \dot{z}}{\partial z} \tag{6}$$

$$C = \frac{\partial F}{\partial u} \tag{7}$$

In this study, we are interested in the mPFC–AM system implicated in motivational behavior. The anatomic definitions for mPFC and AM were the same as we used in the resting-state connectivity analyses (Supplementary Fig. 11a), and the task inputs were defined as the two video-viewing conditions (i.e., IVs and CVs), yielding a full-model structure including 16 parameters (Fig. 8g). We first asked the question of whether human imaging data would produce the same functional architecture revealed with animal physiological techniques by applying the Parametric Empirical Bayesian (PEB) frame[55,56] to the intrinsic model (i.e., the field A). The resultant best model indicates reciprocal excitatory connections between mPFC and AM (Fig. 8h), which is consistent with the results from animal data showing positive feedback in the mPFC–AM loop (Fig. 6). Then we asked the second question of how the video-viewing condition modulates the system by applying PEB to the full-model parameters (i.e., the fields A, B, and C). The resultant best model indicates that the task condition modulates the system primarily by dis-inhibiting mPFC (Fig. 8i).

**Slice electrophysiology procedure.** For the mPFC-to-AM pathway, EMX1-Cre mice received injections of AAV-EF1α-DIO-ChR2-EYFP and AAV-retro-CAG-tdTomato into the mPFC ($n = 8$). Mice were deeply anesthetized with isoflurane (60–90 s) and then rapidly decapitated. Coronal slices containing the PFC or AM were cut in ice-cold solution containing (in mM) 92 NMDG, 20 HEPES, 25 glucose, 30 NaHCO₃, 1.2 NaH₂PO₄, 2.5 KCl, 5 Na-ascorbate, 3 Na-pyruvate, 2 thiourea, 10 MgSO₄, 0.5 CaCl₂, saturated with 95% O2 5% CO₂ (pH 7.3–7.4, ~305 mOsm/kg) and incubated for 3–10 min at 35 °C in the same solution. Slices were allowed to recover for a minimum of 30 min at room temperature in artificial cerebrospinal fluid (ACSF) containing (in mM) 126 NaCl, 2.5 KCl, 1.2 MgCl₂, 2.4 CaCl₂, 1.2 NaH₂PO₄, 21.4 NaHCO₃, 11.1 glucose, 3 Na-pyruvate, 1 Na-ascorbate. Recordings were at 31–35 °C in the same solution which was bath perfused at 1.5-3 ml/min. For whole-cell voltage-clamp recordings, intracellular solution contained (in mM) 120 mM CsMeSO₃, 5 mM NaCl, 10 mM TEA-Cl, 10 mM HEPES 1.1 mM EGTA, 4 mM Mg-ATP, 0.3 mM Na-GTP (pH 7.2–7.3, ~290 mOsm/kg). QX-314 (4 mM) was included in some recordings. Whole-cell recordings were performed in tdTomato-expressing PFC neurons projecting to AM, or in tdTomato-expressing AM neurons projecting to PFC. Differential interference contrast optics were used to patch neurons. For ChR2 experiments, either a 473 nm laser (OEM laser systems, maximum output 500 mW) attached to fiber optic cable or 460 nm LED (CoolLED) connected to the objective was used to deliver light to the slice. Synaptic responses were briefly titrated with light stimulation of ChR2-expressing terminals with varying light intensity in slice recording. Response in 60–100% range of maximum response amplitude (light intensity of 2–12 mW) was used for blocker treatment experiment. For experiments testing the involvement of glutamatergic transmission in the AM-projecting PFC neurons or in the PFC-projecting AM neurons, we measured optogenetically evoked EPSCs (Fig. 6c–e) by giving 2 ms light pulses at a frequency of 0.1 or 0.5 Hz. In cells that showed a light-evoked response, DNQX (20–30 μM) and AP5 (50 μM) were added to the recording bath solution after a stable baseline measurement was recorded. Recordings were discarded if series resistance or input resistance changed >20% throughout the course of the recording. An Axopatch 200B amplifier (Molecular Devices) and Axograph X software (Axograph Scientific) were used to record and collect the data shown in Fig. 6e. A Multiclamp 700B and pCLAMP 11 (Molecular Devices) were used for data shown in Fig. 6d. The data were filtered at 10 kHz and digitized at 4–20 kHz.

*Fiber-photometry calcium-signal recordings with optogenetic stimulation.* We used the fiber-photometry system manufactured by Doric Lenses (Quebec, Canada). The light pulse generator, consisting of a driver and LED units, produced light wavelengths (465 nm and 405 nm) in sinusoidal waveforms (208 Hz and 530 Hz, respectively), which were fed into a fluorescence minicube via patchcables (NA: 0.48; core diameter: 400 μm). The minicube combined these wavelength lights and sent the combined beam into the brain via a patchcable (NA: 0.48; core diameter: 400 μm) connected to the implanted optic fiber (NA: 0.48; core diameter: 200 or 400 μm). The same patchcable/optic-fiber assembly, in turn, carried the emission of GCaMP (525 nm) as well as control (430 nm) back to the minicube, which then separated emission bandpass with beamsplitters and sent them to photoreceiver modules (Newport: model 2151) via patchcables (NA: 0.48; core diameter: 600 μm). The photoreceiver modules quantified signals and sent them to the Fiber Photometry Console, which was controlled by the Doric Neuroscience Studio software and which synchronized the acquisition of the data with the output of the laser stimulus. Recording data were collected at the rate of 1200 Hz.

Med Associates' system (Fairfax, VT) produced the trains (1, 2, 4, or 8 pulses at 25 Hz) delivered in random order on a variable-interval schedule with the mean interval of 15 s by controlling the pulse generator (Doric Lenses) that, in turn, controlled a laser for generating a 3-ms pulse of a 635-nm wavelength. Each mouse received 41 trains for each pulse (41 trains × 4 train types = 164 total trains) at the AM, VStr, and VTA. Half of the mice received the trains at the AM first and then the VStr, while the other half received the trains in the reverse order. All mice received trains at the VTA after being tested at the AM and the VStr. For the delivery of VTA stimulation, the laser was connected to the minicube, which relayed lights to the VTA via the same probe for both the fiber photometry and optogenetic procedures. For mPFC stimulation, the laser was directly connected to the optic-fiber probe targeting the mPFC.

Photometry analysis: We used custom-written MATLAB code to transform and analyze photometry data. We first binned the separate GCaMP and Control channels into one-minute epochs. We transformed the Control channels of each bin to a linear fit of its respective GCaMP channel, and calculated dF/F for each bin with the formula (GCaMP-FittedControl) FittedControl⁻² [79]. We then extracted the dF/F trace corresponding to two seconds before and after every light train, extracted the corresponding area under the curve, and performed a repeated-measures ANOVA with Time (before and after stimulation train) and Pulse (1, 2, 4, and 8 pulses) (GraphPad Prism) on the extracted area under the curve for each region of the individual animal. We performed post hoc *t* tests with Benjamini and Hochberg correction on Time for each pulse when the interaction was significant.

*Histology.* Mice were intracardially perfused with 1× PBS followed by 4% paraformaldehyde. Brains were kept in a 30% sucrose in PBS solution over 1–2 days before cutting. After freezing, brains were coronally sectioned at 40 μm with a

cryostat, mounted directly onto slides, and coverslipped with DAPI nuclear counterstain. Optical fiber placements and fluorophore expression were determined with fluorescent microscopy.

*Statistical analysis.* Behavioral data were analyzed with Statistica v6.1 or GraphPad Prism v4 & v8. Electrophysiological recording data were analyzed with Molecular Devices' Clampfit v10.6 and Origin Pro v9.2 (OriginLab Corporation, MA, USA).

**Reporting summary**. Further information on research design is available in the Nature Research Reporting Summary linked to this article.

## Data availability

The following two types of data were deposited in Zenodo and are available at this link: https://doi.org/10.5281/zenodo.5767725: (1) the behavioral data summarized in Figs. 2c, d, 2f, and 3b and Suppl. Figure 3b, c; (2) the fiber-photometry calcium-signal data summarized in Figs. 7b, c and Suppl. Figure 9. In addition, fMRI activation images supplementing for Figs. 3d and 8 and Supplementary Figs. 5 and 11 were also deposited in Zenodo and are available at these links: rat: https://doi.org/10.5281/zenodo.5794424; and humans: https://doi.org/10.5281/zenodo.5794451. The fMRI raw data have not been made available due to being analyzed for other projects. Source data are provided with this paper.

## Code availability

The MATLAB code used for fiber-photometry data analysis is available at Github/ Zenodo[80]: https://zenodo.org/badge/latestdoi/337120575.

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

## Acknowledgements

The authors thanks National Institute on Drug Abuse (NIDA)Transgenic Breeding Facility for transgenic mice, NIDA Genetic Engineering and Viral Vector Core for viral vectors, and NIDA Histology Core for cell counting. The Intramural Research Program of NIDA supported the present work by providing funds to S.I. and Y.Y. The China Scholarship Council (No. 201306590020) partly supported C.Y. The National Natural Science Foundation of China (No. 81971245) supported Y.H. and C.H.

## Author contributions

Conceptualization: S.I., C.Y. and Y.H.; formal analysis: C.Y., Y.H., R.F.D. and S.I.; investigation: C.Y., Y.H., A.D.T., C.T.P., L.A.R., A.J.K., R.F.D., S.J., A.T., A.F.P., C.N., C.B.C., C.S., Y.A., S.C.L. and J.M.C.; software: R.F.D. and C.M.-A.; writing—original draft: S.I.; writing—review editing: C.Y., Y.H., A.D.T., C.T.P., L.A.R., A.J.K., R.F.D., S.J., A.T., A.F.P., C.N., C.B.C., C.S., C.M.-A., D.V.W, H.L., Y.Y., Y.A., S.C.L., J.M.C. and S.I.; supervision: H.L., D.V.W., Y.Y. and S.I.; project Administration: S.I.

## Funding

## Competing interests

The authors declare no competing interests.
