## [Peer Review File · Nature Communications]

Medial prefrontal cortex and anteromedial thalamus
interaction regulates goal-directed behavior and dopaminergic
neuron activityREVIEWER COMMENTS

Reviewer #1 (Remarks to the Author):

This paper presents very interesting evidence that rodent medial prefrontal cortex (mPFC) supports optogenetic intracranial self-stimulation (ICSS) and that stimulation of the anteromedial thalamic nucleus (AM) affects this function while stimulating dopamine release by ventral tegmental area neurons. The authors provide convincing evidence that stimulation of dorsal and ventral mPFC (but not dorsal motor areas) support high rates of lever press responding that are short lived. The increase was specific for the active (vs inactive) lever, reversed when active and inactive levers were switched, responded to extinction and re-acquisition, depended on frequency of stimulation, and was not observed for controls lacking CHR2 expression.

The implication of mPFC in reinforcement is consistent with evidence that rodent mPFC contains populations of neurons that are excited during positive reinforcement in different behavioral tasks and that lesions or inactivation of mPFC have broad effects on reinforcement-guided learning and decision-making. Similar data, providing convergent evidence for a role in motivation or reinforcement, are lacking for AM. This raises the question of whether AM has a direct role in motivation or reinforcement or if the effects of AM ICSS reported here are the result of thalamic stimulation indirectly activating mPFC (producing a pattern of activation comparable to direct mPFC ICSS).

It is well established that AM is driven by afferents that originate throughout mPFC, however, retrograde tracing studies indicate that AM projects heavily to anterior cingulate and less heavily to medial agranular (motor) areas, with few if any projections to more ventral areas of PL and IL (Hoover & Vertes Brain Struct Func 212:149-179). Could labeling observed in PL (Fig 5,6) reflect the spread of thalamic injections into adjacent areas of MD (which has robust projections to all areas of mPFC)?

How confident can we be that the positioning of optic fibers differentiate between the effects of activating different thalamic nuclei? To me the optic fiber shown in Fig 4B appears positioned to activate lateral Re and a very limited area of medial ventral AM. The maps of optic fiber tips shown in Fig 4C show a dense concentration around AM with relatively few specifically targeting MD or Re. This is not surprising given the focus of the paper on AM, however, it raises questions about the validity of using these data to argue that MD and Re lack ICSS effects comparable to AM. It would be more convincing to have equal numbers of probes specifically targeting different nuclei (i.e. experimentally manipulated), carefully positioned to avoid light paths activating multiple nuclei. I don't think that these data are sufficient to support the argument that the effects of thalamic ICSS are localized to AM and do not involve adjacent midline, mediodorsal, or maybe even intralaminar nuclei.

The human fMRI study shows an interesting parallel to the rodent studies. However, they raise the same issue of the uncertainty localizing thalamic activation to AM. Much of the problem is inherent in the anatomy: central thalamus contains many functionally distinct nuclei that are tortuously juxtaposed in a limited space. This paper presents interesting evidence that mPFC and thalamus interact to support ICSS. It would be improved by a more critical discussion of the localization of this function in thalamus.

RG Mair

Reviewer #2 (Remarks to the Author):

Yang et al. report an exciting and well conducted series of studies examining the interactions between the medial prefrontal cortex and the anteromedial thalamus in aspects of appetitive motivation and activity of dopamine neurons. This is an ambitious manuscript, reporting results from mice, rats, and humans. The experiments were well designed, conducted, and reported. The appropriate controls

have, in most cases, been included, and the conclusion – that the corticothalamic loop studied here contributes to motivation, reinforcement, and dopamine neuron activity, is a novel and, I think important one. These findings will be of broad interest to several fields.

I have no substantive criticisms regarding the design, approach, or interpretation. But I do have some comments for the authors consideration.

1. After going to much trouble to probe the effective sites in the PFC from which ICSS could be obtained, the remainder of the manuscript focussing on pathways, refers only to 'mPFC'. I found this a little confusing. Was it IL? PL? both? More generally, I think readers would benefit from some guidance in how to think about the PFC region being studied in Figure 2 onwards and I also suggest some attempt to link to the suggestions of Laubach regarding mPFC nomenclature.

2. I think the authors should be commended for chasing down the collaterals of the mPFC neuron projecting to the AM (Figure 5; see also p. 7). This is an important experiment because it helps readers understand the effects of mPFC stimulation better as well as better appreciate the neural networks being recruited. However, as far as I can tell, this is based on one mouse with successful/correct viral expression. This does raise concerns about reliability and Figure 5 itself is of lower quality than the remaining data presented. I do think readers would be more persuaded by additional animals and higher quality data here.

3. The effects on VTA DA neurons were very interesting – and these were heroic experiments. But, I was not sure what exactly the authors meant by “as long as the optic fibre placements...were optimal”. Which placements exactly? All of the AM placements in Supp Fig 8? Or only some?

a. Related to this, there are no error bars on the photometry data in Supp Fig 7 or Figure 7 and the duration of the traces varies between the main figure 7 (+/- 2s) and the Suppl Fig 7. Why are different durations reported?

b. What were the post-hoc t-tests on time, exactly? -1 to 0 s versus 0 - 1 s?

4. I enjoyed the human fMRI experiment at the end (even if it seemed a stretch), but I am not expert in human fMRI.

5. Supp Figure 2 – I see what you are trying to achieve but I am not sure a figure counts as a 'theoretical model' – without formal modelling I am not sure what the reader gains. This could be unpacked a little more or presented as a descriptive model.

6. Readers may benefit from better understanding what is depicted, exactly, in histological reconstructions of AAV expression. Are these maximal spread? Maximal and minimum? And should readers be concerned with the extent of this spread across large parts of the thalamus and PFC?

7. There are some consistent typographical errors such as:

a. I believe Re thalamus it is reuniens not reunions,

b. Signed not singed etc in the methods

Reviewer #3 (Remarks to the Author):

In this manuscript, Yang et al. demonstrated a novel cortico-thalamic loop regulating motivated behavior and dopaminergic activity. Firstly, by combining optogenetics with an intracranial self-stimulation behavior (ICSS) paradigm, the authors found in mice that among the PFC subregions, stimulation of the mPFC supported high rates of ICSS and such stimulation-induced motivation is short-lasting. Secondly, fMRI imaging on anesthetized rats revealed that mPFC stimulation activated many downstream regions. ICSS can be induced by stimulating mPFC axonal terminals in these downstream regions, especially the anteromedial thalamic nucleus (AM). Thirdly, by conducting tracing experiment, optogenetics and fiber photometry in mice, the authors identified that the mPFC-

AM circuit form a positive feedback loop, which supported high rate of ICSS and also modulated the activity of midbrain dopaminergic neurons. Finally, the fMRI study on humans viewing video-clips revealed that mPFC and AM were activated simultaneously when watching reinforcing videos compared with control videos. Based on these results, the authors proposed that the mPFC-AM circuit regulates motivation and dopaminergic neuron activity. These findings provide a novel understanding on the motivation circuit which may provide insights to guide future studies. I am enthusiastic for its publication in Nature Communication after addressing a few issues below.

Major issues:

1. The authors suggest mPFC-AM circuit to be important for both modulation of motivation and modulation of dopamine activity, but without providing a link between the two. As previous studies reported that optogenetic stimulation of dopamine neurons in the VTA induced ICSS (Witten et al., 2011, Neuron), it should be relevant to figure out whether mPFC-AM circuit modulates motivation through its modulation of dopaminergic neuron activity, for example, by examining mPFC-AM-stimulation-induced ICSS when VTA is inhibited.
2. By performing fMRI experiment during mPFC stimulation in rats, the authors identified stimulation-evoked BOLD signals in the downstream regions of mPFC, including the anterior insular area (AI), ventral striatum (VStr), anterior thalamus (ATh), and hypothalamus were significantly correlated with ICSS levels (Fig 3e). However, in the following figure (Fig 4), the authors suddenly brought up the medial dorsal striatum (mDStr) and internal capsule (IC), which were not mentioned in the fMRI results. Please clarify the logical links between Fig 3 and 4. In addition, the authors stated that they further dissected the mPFC-IC pathway because "IC carries cortical information to many downstream regions (page 7, paragraph 1)" without any reference. Please provide the evidence to prove that mPFC project to downstream regions through IC.
3. In Fig. 6c and 6d, by conducting brain slice electrophysiology experiments, the authors found a positive-feedback organization between the mPFC and the AM. The stimulation of mPFC-to-AM neurons excites AM-to-mPFC neurons and vice versa. Remarkably, the application of glutamate receptor antagonists did not fully block the evoked-EPSC. Please provide an explanation.

Minor issues:

1. In Fig.3e, according to the figure legend, n = 10 rats received fMRI scanning during photostimulation of mPFC. However, only n=9 data points were presented for correlations between ICSS responses and the levels of BOLD signals in the AI. Such inconsistency was also found in Fig.7b. It is stated in figure legend that "Colored lines indicate data of 8 mice", however, only the data of 6 mice was shown. Please clarify how data were selected in both cases.
2. Fig. 5 showed that mPFC neurons projecting to the AM have collateral projections to the VStr, VTA or both. The authors conducted this tracing experiment in n=3 mice (Page 7, 2nd paragraph), however, only one representative case was showed in Fig.5. Do the remaining 2 mice show consistent collateral projecting pattern as the representative one?
3. Typos were found in the sentences "We found that the mice had significantly higher rates of ICSS rates o for trains with 25 or 50 Hz..." (page 5, paragraph 2) and "We obtained a separate singed consent-form We acquired individualized video clips (IVs; 6min long) by singing in the account of ..." (page 17, paragraph 1). Please check.

Reference

Witten, I. B., Steinberg, E. E., Lee, S. Y., Davidson, T. J., Zalocusky, K. A., Brodsky, M., Yizhar, O., Cho, S. L., Gong, S., Ramakrishnan, C., Stuber, G. D., Tye, K. M., Janak, P. H., & Deisseroth, K. (2011). Recombinase-driver rat lines: tools, techniques, and optogenetic application to dopamine-mediated reinforcement. *Neuron*, 72(5), 721–733. <https://doi.org/10.1016/j.neuron.2011.10.028>

Hailan Hu

Reviewer #4 (Remarks to the Author):

The authors did conduct an impressive amount of work to investigate neural circuits involved in motivational regulation for action. They combined optogenetics in mice with fMRI in rats and humans.

Among the targeted regions in their optogenetic experiment in mice, areas of the medial frontal cortex were the regions associated with stronger behavioural effects.

The authors acknowledged the of the anatomical heterogeneity of this region but I failed to understand the rationale behind their subdivisions. Some regions called orbital were included in their mpfc cluster despite the authors considering an orbital ROI, a cingular region was considered a motor region.

Then the authors decided to focus on a subdivision of the mpfc, the infralimbic area although behavioural effects appear to be similar if not stronger in other regions located on the medial wall (PL, Cg2, MO, VO).

The authors conducted a two-way anova with only two levels for the number of areas and only 5 levels for the number of sessions. If they focused their analysis on areas located on the midline, why did they not conduct a 6 areas x 10 sessions anova?

To further characterize the link between midline stimulation and behaviour, the authors introduced one reversal, with the new contingencies being presented for two sessions. Was there a difference between the different stimulated sites? The authors mentioned in the methods section that other tests were conducted. Were the different stimulation sites pulled together or were the effects more specific for IL?

Secondly, they manipulated the cost of the task by increasing the number of lever presses for getting a stimulation. While the other measures were taken over 30min, for this specific condition authors were only considering a 5min window. However, in the methods section, authors mentioned a change every 10min. Did I misunderstand how they conducted their manipulation? Was the order of the switches done randomly?

The authors then investigated whether a similar effect could be observed in rats prior to an fMRI study. The rats did press although at a lower rate than the mice. Could it be due to the stimulation parameters? Based on the mice data (fig 2f) they were not optimal. Increasing the cost of a stimulation (using a time manipulation rather than an action contingency one) did also reduce the number of lever presses.

A stimulation of the right IL, elicited changes in several areas including midline structures but those BOLD changes were not correlated with lever presses. Were the effects measured ipsilaterally to the stimulated IL or did the authors averaged the BOLD signals over both hemispheres?

Having looked at the effects of medial wall stimulations, the authors investigated the efficacy of stimulations of terminations of those regions located notably in the ventral striatum and the thalamus. Both were associated with strong behavioural effects. Weren't the effects more consistent and stronger in the striatum than in the AM? While interactions between AM and the medial are less investigated than interactions striato-cingulate, the authors should nevertheless acknowledge the importance of the later pathway.

In their last study the authors did conduct an fMRI study in humans. Observed effects are not located in their preferred ROI, IL, which would correspond to the subgenual cingulate cortex (area25) in humans (see for instance Schaeffer et al. PNAS 2020; Heilbronner et al. 2014 Biological Psychiatry). This discrepancy with results obtained in rodents might be related to the fact that the stronger effects were not in IL in rodents. Could the key area in the rodent studies be PL? Furthermore, the localized effect identified here speaks again for a functional heterogeneity of the medial wall.

Reviewer Comments/Author Responses

Reviewer #1 (Remarks to the Author):

This paper presents very interesting evidence that rodent medial prefrontal cortex (mPFC) supports optogenetic intracranial self-stimulation (ICSS) and that stimulation of the anteromedial thalamic nucleus (AM) affects this function while stimulating dopamine release by ventral tegmental area neurons. The authors provide convincing evidence that stimulation of dorsal and ventral mPFC (but not dorsal motor areas) support high rates of lever press responding that are short lived. The increase was specific for the active (vs inactive) lever, reversed when active and inactive levers were switched, responded to extinction and re-acquisition, depended on frequency of stimulation, and was not observed for controls lacking CHR2 expression.

The implication of mPFC in reinforcement is consistent with evidence that rodent mPFC contains populations of neurons that are excited during positive reinforcement in different behavioral tasks and that lesions or inactivation of mPFC have broad effects on reinforcement-guided learning and decision-making. Similar data, providing convergent evidence for a role in motivation or reinforcement, are lacking for AM. This raises the question of whether AM has a direct role in motivation or reinforcement or if the effects of AM ICSS reported here are the result of thalamic stimulation indirectly activating mPFC (producing a pattern of activation comparable to direct mPFC ICSS).

The authors' response:

We believe that motivation and reinforcement arise from the activity of neural networks consisting of extended regions. Consistently, neurons can communicate with each other on a sub-second scale, while motivated behavior takes over seconds and minutes. Accordingly, the AM is a component of such neural network, which includes mPFC neurons and VTA dopaminergic neurons. Therefore, it is more adequate to state that the AM interact with the mPFC in goal-directed motivation than that the AM is involved indirectly in goal-directed motivation via the mPFC. We have modified the 1st paragraph of the discussion section (pp. 10-11) to emphasize this point.

It is well established that AM is driven by afferents that originate throughout mPFC, however, retrograde tracing studies indicate that AM projects heavily to anterior cingulate and less heavily to medial agranular (motor) areas, with few if any projections to more ventral areas of PL and IL (Hoover & Vertes Brain Struct Func 212:149-179). Could labeling observed in PL (Fig 5,6) reflect the spread of thalamic injections into adjacent areas of MD (which has robust projections to all areas of mPFC)?

The authors' response:

We appreciate that this is a critical comment. While MD neurons may project to the mPFC more robustly than AM neurons, AM neurons do project to the mPFC. First, van Groen et al. (1999, Brain Res Rev 30:1-26) reported that the AM projects to the mPFC, including the prelimbic and medial orbital cortex, as shown below.

In addition, our own data provide evidence. First, we observed and recorded neurons that were retrogradely-labeled by the injection of AAV-retro-tdTomato into the mPFC (data shown in Fig. 6e). Second, we now add photomicrograms showing Cre-dependent tdTomato expressions for which Cre was retrogradely expressed by injecting into the mPFC, shown below. These photomicrograms are from the fiber photometry experiment described in Fig. 7c. The previous manuscript displayed stimulation sites on coronal drawings, and they are now replaced with these (Suppl. Fig. 10b). As shown, the AM contains both retrogradely-expressed EYFP (Cre) and Cre-dependent mCherry.

Third, retrogradely-labeled by injection of AAV-ChR2-EYFP into the mPFC supported ICSS at the AM as shown in Fig. 6c. The stimulation sites are shown below (Suppl. Fig. 8) and Fig. 6c. Unfortunately, this AAV vector did not express EYFP to reveal the shape of cells that contained it; that is, it produced smeared EYFP expressions. As a result, it was hard to detect soma defined by EYFP-labels. Moreover, photostimulation appeared to have bleached the affected area of EYFP expression. Nevertheless, the fact that the mice display vigorous ICSS at the AM supports that AM neurons project to the mPFC.

How confident can we be that the positioning of optic fibers differentiate between the effects of activating different thalamic nuclei? To me the optic fiber shown in Fig 4B appears positioned to activate lateral Re and a very limited area of medial ventral AM. The maps of optic fiber tips shown in Fig 4C show a dense concentration around AM with relatively few specifically targeting MD or Re. This is not surprising given the focus of the paper on AM, however, it raises questions about the validity of using these data to argue that MD and Re lack ICSS effects comparable to AM. It would be more convincing to have equal numbers of probes specifically targeting different nuclei (i.e. experimentally manipulated), carefully positioned to avoid light paths activating multiple nuclei. I don't think that these data are sufficient to support the argument that the effects of thalamic ICSS are localized to AM and do not involve adjacent midline, mediodorsal, or maybe even intralaminar nuclei.

The authors' response:

We appreciate that this is another critical point. We have added new data involving ICSS with mPFC-to-MD (n = 9), mPFC-to-Re (n = 6), mPFC-to-AM (n = 3), and vicinity (n = 4) and updated Figure 4. For your convenience, the stimulation sites of new mice are indicated by red arrows, as shown below. Although the stimulation of mPFC-to-MD or mPFC-to-Re neurons supported ICSS in some mice, the rates of ICSS at these sites are generally lower than we observed for the AM (Figure 4d), suggesting that the AM is more important in goal-directed motivation than the MD or Re.

In addition, we have examined the effects of VTA inhibition on ICSS reinforced by the stimulation of mPFC-to-AM pathway in 4 mice, described in Fig. 7e. The stimulation sites of these mice are clearly indicated in coronal sections shown below and in Suppl. Fig. 10c.

Moreover, we point out two observations that support the role of the AM in ICSS. As discussed above, selective stimulation of AM neurons projecting to the mPFC supported vigorous ICSS (Fig. 6c; Suppl. Fig. 8). Second, our slice electrophysiology experiment showed that the stimulation of mPFC-to-AM neurons activates AM neurons (Figure Fig. 6d).

The human fMRI study shows an interesting parallel to the rodent studies. However, they raise the same issue of the uncertainty localizing thalamic activation to AM. Much of the problem is inherent in the anatomy: central thalamus contains many functionally distinct nuclei that are tortuously juxtaposed in a limited space. This paper presents interesting evidence that mPFC and thalamus interact to support ICSS. It would be improved by a more critical discussion of the localization of this function in thalamus.

The authors' response:

Thanks for this insightful comment. We agree that dissecting functional specificity of thalamic nuclei is challenging due to the compacted anatomic space and complex cortical afferent inputs. Combing our own results and existing findings, we have discussed in more details about the functional localization of ICSS in thalamus. See the 2nd paragraph on page 12.

RG Mair

Reviewer #2 (Remarks to the Author):

Yang et al. report an exciting and well conducted series of studies examining the interactions between the medial prefrontal cortex and the anteromedial thalamus in aspects of appetitive motivation and activity of dopamine neurons. This is an ambitious manuscript, reporting results from mice, rats, and humans. The experiments were well designed, conducted, and reported. The appropriate controls have, in most cases, been included, and the conclusion – that the corticothalamic loop studied here contributes to motivation, reinforcement, and dopamine neuron activity, is a novel and, I think important one. These findings will be of broad interest to several fields.

I have no substantive criticisms regarding the design, approach, or interpretation. But I do have some comments for the authors consideration.

The authors' response:

Thank you. We appreciate your positive general assessment.

1. After going to much trouble to probe the effective sites in the PFC from which ICSS could be obtained, the remainder of the manuscript focussing on pathways, refers only to 'mPFC'. I found this a little confusing. Was it IL? PL? both? More generally, I think readers would benefit from some guidance in how to think about the PFC region being studied in Figure 2 onwards and I

also suggest some attempt to link to the suggestions of Laubach regarding mPFC nomenclature.
The authors' response:

Thanks for the point. First, we made sure that the progression of the study is clearly discussed. We now state “Because the stimulation of the IL supported quantitatively similar levels of ICSS as that of the other mPFC sub-regions, these sub-regions of the mPFC are grouped for the rest of the study”. See the last sentence of the second paragraph on page 4. In addition, we have added Supplementary Note 1 to discuss the areas of the PFC, specifically how the sequence of work overtime could cause confusions (p. 2 of Suppl. Materials).

2. I think the authors should be commended for chasing down the collaterals of the mPFC neuron projecting to the AM (Figure 5; see also p. 7). This is an important experiment because it helps readers understand the effects of mPFC stimulation better as well as better appreciate the neural networks being recruited. However, as far as I can tell, this is based on one mouse with successful/correct viral expression. This does raise concerns about reliability and Figure 5 itself is of lower quality than the remaining data presented. I do think readers would be more persuaded by additional animals and higher quality data here.

The authors' response:

We appreciate recognizing possible value of the work. We have added additional mice (n = 3) whose data are described in Figure 5 and Suppl. Figs. 6 and 7. We also made methodological improvements in two ways. In the original experiment, we used an AAV-retro-mCherry-Cre for AM injections, and this did not allow us to distinguish between injection sites and AM's neighboring regions, which project to the AM, because of retrograde expressions of mCherry. The new mice (n = 3) received intra-AM delivery of AAV-retro-eYFP-Cre, which does not interfere detecting mRuby, the marker of mPFC terminal boutons. In addition, intra-AM delivery of AAV-retro-eYFP-Cre were performed with the **silk fibroin method** (Jackman, S. L. et al. Silk Fibroin Films Facilitate Single-Step Targeted Expression of Optogenetic Proteins. Cell Rep 22, 3351-3361, doi:10.1016/j.celrep.2018.02.081 (2018)), which resulted in confined delivery of the AAV to the AM, minimizing the diffusion of injected vectors. With these new procedures, we were able to determine the delivery sites of AAV-retro-eYFP-Cre. The deliveries were selective to the AM because we only observed mRuby expressed in the AM, but not in the MD or the reunions. We are confident that the results of the new experiment confirmed that the AAV-retro-Cre was taken up by mPFC neurons projecting to the AM and that AM projecting mPFC neurons have collateral projections to the VStr, VTA, or both. While our original data are no longer shown due to poor quality of photomicrograms, better new data are shown in Figure 5 and Suppl. Figs. 6 and 7.

3. The effects on VTA DA neurons were very interesting – and these were heroic experiments. But, I was not sure what exactly the authors meant by “as long as the optic fibre placements...were optimal”. Which placements exactly? All of the AM placements in Supp Fig 8? Or only some?

The authors' response:

By that, we meant that the recorded signals depended on several factors coming together: the locations of the tips of stimulation and recording probes in relation to the target regions and

expression levels opsins at the target regions. When any of one of these conditions were somewhat compromised, signals were lower. This notion explains the differential signal levels we obtained. Larger signals were obtained from mice MNV3 and 4, which had good opsin expressions and probe placements, while MNV1, 2, 5 and 7, which had good opsin expressions, but probe placements were within reasonable distance from the targets, but not at them (Suppl. Fig. 9). We have re-written the paragraph. See the first full paragraph on page 9.

a. Related to this, there are no error bars on the photometry data in Supp Fig 7 or Figure 7 and the duration of the traces varies between the main figure 7 (+/- 2s) and the Suppl Fig 7. Why are different durations reported?

The authors' response:

Errors are now shown in the data displayed in Suppl. Fig. 9 (formerly Suppl. Fig. 7). Note that errors are not shown for Fig. 7, to avoid making the line graphs too busy. Suppl. Fig. 9 provides errors for the data shown in Fig. 7. In addition, consistent with that of Fig. 7, Suppl. Fig. 9 now shows data for the +/- 2-s duration.

b. What were the post-hoc t-tests on time, exactly? -1 to 0 s versus 0 - 1 s?

The authors' response:

t-Tests compared data between the 2-s period prior and the 2-s period after the onset of photostimulation. This is now noted in the Suppl. Fig. 9 legend. Thanks.

4. I enjoyed the human fMRI experiment at the end (even if it seemed a stretch), but I am not expert in human fMRI.

The authors' response:

Thanks for your positive take on the experiment.

5. Supp Figure 2 – I see what you are trying to achieve but I am not sure a figure counts as a ‘theoretical model’ – without formal modelling I am not sure what the reader gains. This could be unpacked a little more or presented as a descriptive model.

The authors' response:

We have replaced “theoretical” with “conceptual” for the legend, implying that the models are descriptive in nature. Thanks.

6. Readers may benefit from better understanding what is depicted, exactly, in histological reconstructions of AAV expression. Are these maximal spread? Maximal and minimum? And should readers be concerned with the extent of this spread across large parts of the thalamus and PFC?

The authors' response:

We now clearly describe that those sections depict the spread of AAVs for each injection. See the legend of Suppl. Figs. 8.

7. There are some consistent typographical errors such as:

- a. I believe Re thalamus it is reuniens not reunions,
- b. Signed not singed etc in the methods

The authors' response:

Thanks. We have corrected them.

Reviewer #3 (Remarks to the Author):

In this manuscript, Yang et al. demonstrated a novel cortico-thalamic loop regulating motivated behavior and dopaminergic activity. Firstly, by combining optogenetics with an intracranial self-stimulation behavior (ICSS) paradigm, the authors found in mice that among the PFC subregions, stimulation of the mPFC supported high rates of ICSS and such stimulation-induced motivation is short-lasting. Secondly, fMRI imaging on anesthetized rats revealed that mPFC stimulation activated many downstream regions. ICSS can be induced by stimulating mPFC axonal terminals in these downstream regions, especially the anteromedial thalamic nucleus (AM). Thirdly, by conducting tracing experiment, optogenetics and fiber photometry in mice, the authors identified that the mPFC-AM circuit form a positive feedback loop, which supported high rate of ICSS and also modulated the activity of midbrain dopaminergic neurons. Finally, the fMRI study on humans viewing video-clips revealed that mPFC and AM were activated simultaneously when watching reinforcing videos compared with control videos. Based on these results, the authors proposed that the mPFC-AM circuit regulates motivation and dopaminergic neuron activity. These findings provide a novel understanding on the motivation circuit which may provide insights to guide future studies. I am enthusiastic for its publication in Nature Communication after addressing a few issues below.

The authors' response:

Thank you. We appreciate your support for the study.

Major issues:

1. The authors suggest mPFC-AM circuit to be important for both modulation of motivation and modulation of dopamine activity, but without providing a link between the two. As previous studies reported that optogenetic stimulation of dopamine neurons in the VTA induced ICSS (Witten et al., 2011, Neuron), it should be relevant to figure out whether mPFC-AM circuit modulates motivation through its modulation of dopaminergic neuron activity, for example, by examining mPFC-AM-stimulation-induced ICSS when VTA is inhibited.

The authors' response:

We have added two new experiments to examine effects of the inhibition of VTA dopamine neurons on ICSS reinforced by the stimulation of the mPFC-to-AM or the AM-to-mPFC pathways. We used TH-Cre mice that had expressed hM4Di in VTA DA neurons. The inhibition of VTA dopamine neurons significantly decreased both rates of ICSS reinforced by the stimulation of the mPFC-to-AM (Fig. 7e) and the AM-to-mPFC (Fig. 7g) pathways. In addition, we examined effects of systemic injection of the dopamine receptor antagonist SCH on them and found the treatment significantly decreased ICSS rates (Fig. 7f, h).

2. By performing fMRI experiment during mPFC stimulation in rats, the authors identified stimulation-evoked BOLD signals in the downstream regions of mPFC, including the anterior insular area (AI), ventral striatum (VStr), anterior thalamus (ATh), and hypothalamus were significantly correlated with ICSS levels (Fig 3e). However, in the following figure (Fig 4), the authors suddenly brought up the medial dorsal striatum (mDStr) and internal capsule (IC), which were not mentioned in the fMRI results. Please clarify the logical links between Fig 3 and 4. In addition, the authors stated that they further dissected the mPFC-IC pathway because “IC carries cortical information to many downstream regions (page 7, paragraph 1)” without any reference. Please provide the evidence to prove that mPFC project to downstream regions through IC.

The authors’ response:

Thanks for detecting the lack of logical flow in the narrative. We now clearly describe that mPFC stimulation increased BOLD signals in the mDStr and IC (1st paragraph on page 7) and that this observation led to the examination of the mDStr and IC (2nd paragraph on page 7). In addition, we now provide a reference for the IC and explain how the mDStr and IC are relevant for the experiment (2nd paragraph on page 7).

3. In Fig. 6c and 6d, by conducting brain slice electrophysiology experiments, the authors found a positive-feedback organization between the mPFC and the AM. The stimulation of mPFC-to-AM neurons excites AM-to-mPFC neurons and vice versa. Remarkably, the application of glutamate receptor antagonists did not fully block the evoked-EPSC. Please provide an explanation.

The authors’ response:

AAV vectors are found to have retrograde properties. We suspected the following: The injection of AAV-ChR2 into the mPFC resulted in not only anterograde expression of opsins, but also retrograde expression; therefore, AM neurons may have expressed ChR2 via a retrograde mechanism, and photostimulation of the AM directly excited AM neurons; because ChR2 mediated EPSCs, glutamate receptor antagonists had no effect. To address this, we used Emx1-Cre mice to selectively express ChR2 in cortical neurons; emx1 is selectively expressed in cortical neurons, but not AM neurons. EPSC in AM neurons evoked by mPFC terminal stimulation were clearly blocked by glutamate receptor antagonists (Fig. 6d).

Minor issues:

1. In Fig.3e, according to the figure legend, n = 10 rats received fMRI scanning during photostimulation of mPFC. However, only n=9 data points were presented for correlations between ICSS responses and the levels of BOLD signals in the AI. Such inconsistency was also found in Fig.7b. It is stated in figure legend that “Colored lines indicate data of 8 mice”, however, only the data of 6 mice was shown. Please clarify how data were selected in both cases.

The authors’ response:

Thanks for detecting the inconsistencies. As for Fig. 3e, the correlation analysis was based on 10 rats. However, one of the rats had a negative value for BOLD signal, resulting in an omission of the data point. Note that negative BOLD signals around zero merely indicate a lack of activation

and that signals can be negative because of noise. We have adjusted the y-axis in such a way that the data point is now shown in Fig. 3e.

As for Fig. 7b, the experiment involved 6, but not 8, mice. We have corrected the typo.

2. Fig. 5 showed that mPFC neurons projecting to the AM have collateral projections to the VStr, VTA or both. The authors conducted this tracing experiment in n=3 mice (Page 7, 2nd paragraph), however, only one representative case was showed in Fig.5. Do the remaining 2 mice show consistent collateral projecting pattern as the representative one?

The authors' response:

We have added new data of 3 mice for this experiment, and the results of the three mice are shown in Fig. 5 and Supplementary Figs. 6 and 7. The results of new mice are consistent with those we reported previously. However, because of poor photo quality, previous data are no longer shown.

3. Typos were found in the sentences “We found that the mice had significantly higher rates of ICSS rates o for trains with 25 or 50 Hz...” (page 5, paragraph 2) and “We obtained a separate singed consent-form We acquired individualized video clips (IVs; 6min long) by singing in the account of ...” (page 17, paragraph 1). Please check.

The authors' response:

We have corrected typos. Thank you.

Reference

Witten, I. B., Steinberg, E. E., Lee, S. Y., Davidson, T. J., Zalocusky, K. A., Brodsky, M., Yizhar, O., Cho, S. L., Gong, S., Ramakrishnan, C., Stuber, G. D., Tye, K. M., Janak, P. H., & Deisseroth, K. (2011). Recombinase-driver rat lines: tools, techniques, and optogenetic application to dopamine-mediated reinforcement. *Neuron*, 72(5), 721–733. <https://doi.org/10.1016/j.neuron.2011.10.028>

Hailan Hu

Reviewer #4 (Remarks to the Author):

The authors did conduct an impressive amount of work to investigate neural circuits involved in motivational regulation for action. They combined optogenetics in mice with fMRI in rats and humans.

Among the targeted regions in their optogenetic experiment in mice, areas of the medial frontal cortex were the regions associated with stronger behavioural effects.

The authors acknowledged the of the anatomical heterogeneity of this region but I failed to understand the rationale behind their subdivisions. Some regions called orbital were included in

their mpfc cluster despite the authors considering an orbital ROI, a cingular region was considered a motor region.

The authors' response:

Thanks for pointing out the difficulty of understand the areas of the PFC. We now discuss the issue of how the PFC areas are divided in Supplementary Note 1, as follows:

Brain structures are often identified and named without knowing their functions or connectivity. As a result, structural names could cause confusions when subsequent findings begin to suggest their functional roles. One of such structures is the orbital cortex. This label is not based on their function or connectivity, but their location - above the eye sockets, i.e., "orbital". Therefore, while the medial, ventral, and lateral orbital regions share this label, they may not necessarily be grouped together. Indeed, subsequent connectivity work suggested that the medial and ventral orbital regions belong to the medial division of the PFC, while the lateral and dorsolateral orbital regions belong to the lateral division of the PFC, which also include the anterior insular area ^{1,2}. The present study denotes the medial and lateral divisions of the PFC as the medial PFC and orbital PFC, respectively. Similarly, the labels: the area 1 (dorsal part) and area 2 (ventral part) of the cingulate cortex in rodents respectively belong to two different divisions: the motor PFC and the medial PFC ^{1,2}.

References:

- 1 Öngür D, Price JL (2000) The organization of networks within the orbital and medial prefrontal cortex of rats, monkeys and humans. *Cerebral Cortex* 10:206-219.
- 2 Uylings HBM, Groenewegen HJ, Kolb B (2003) Do rats have a prefrontal cortex? *Behavioural Brain Research* 146:3-17.

Then the authors decided to focus on a subdivision of the mpfc, the infralimbic area although behavioural effects appear to be similar if not stronger in other regions located on the medial wall (PL, Cg2, MO, VO).

The authors conducted a two-way anova with only two levels for the number of areas and only 5 levels for the number of sessions. If they focused their analysis on areas located on the midline, why did they not conduct a 6 areas x 10 sessions anova?

The authors' response:

We need to clarify. As discussed in the first paragraph of the result section, our first question was whether the IL is different from the rest of the mPFC with respect to goal-directed motivation. Our results suggested that the answer was no, indicated by the rate of ICSS. Therefore, we stopped making distinction among the sub-regions of the mPFC and considered all mPFC sub-regions as a single group with respect to goal-directed motivation. We now clearly communicate this point in the 2nd paragraph of page 4. Then, we asked our second question of whether the mPFC is different from other neighboring areas. To get at this question, we focused, we performed the 5^{area} x 5^{session} ANOVA for data during the acquisition phase of ICSS. ICSS during the first two sessions did not differ between the 5 areas (statistical results are now shown in the last paragraph of page 4).

To further characterize the link between midline stimulation and behaviour, the authors

introduced one reversal, with the new contingencies being presented for two sessions. Was there a difference between the different stimulated sites? The authors mentioned in the methods section that other tests were conducted. Were the different stimulation sites pulled together or were the effects more specific for IL?

The authors' response:

The reversal test was conducted with the stimulation of the IL and other mPFC sites pooled together, and we did not perform the reversal test in other cortical areas. As alluded above, we now clearly state that the IL and the rest of the mPFC are considered as a unit in the last paragraph of page 4.

Secondly, they manipulated the cost of the task by increasing the number of lever presses for getting a stimulation. While the other measures were taken over 30min, for this specific condition authors were only considering a 5min window. However, in the methods section, authors mentioned a change every 10min. Did I misunderstand how they conducted their manipulation? Was the order of the switches done randomly?

The authors' response:

In the method section on page 14, we noted that each frequency's effects were examined in a 10-min period (or 2 bins of 5 min); the number of presses during the second bin was used to represent the effect of the specific frequency. We now mention this in the figure legend (Fig. 2e) as well.

The authors then investigated whether a similar effect could be observed in rats prior to an fMRI study. The rats did press although at a lower rate than the mice. Could it be due to the stimulation parameters? Based on the mice data (fig 2f) they were not optimal. Increasing the cost of a stimulation (using a time manipulation rather than an action contingency one) did also reduce the number of lever presses.

The authors' response:

Thanks for sharing your thoughts. It is possible that stimulation parameters may not have been optimal for rats. An obvious difference between rats and mice is the body-brain size. Roughly, rats are ten times larger than mice. Photostimulation in rats may have affected smaller areas relative to that of mice whose brains are smaller. Therefore, the rat experiment may have needed to use a greater power of photostimulation than that of mice or multiple probes for better comparison. We now mention this in text (3rd paragraph on p. 6).

A stimulation of the right IL, elicited changes in several areas including midline structures but those BOLD changes were not correlated with lever presses. Were the effects measured ipsilaterally to the stimulated IL or did the authors averaged the BOLD signals over both hemispheres?

The authors' response:

The effects of stimulation on the mPFC were estimated from both left and right hemispheres. The lack of correlation between BOLD changes in the stimulation site (i.e. mPFC) and lever presses may be explained by the ceiling effect of BOLD changes, which failed to reflect the real extent of neural activation. As shown in the bar graph (red bar) on the right, the mPFC showed much higher BOLD activation than any other regions that we measured. We now discuss this issue in the manuscript (1st paragraph on page 7).

Having looked at the effects of medial wall stimulations, the authors investigated the efficacy of stimulations of terminations of those regions located notably in the ventral striatum and the thalamus. Both were associated with strong behavioural effects. Weren't the effects more consistent and stronger in the striatum than in the AM? While interactions between AM and the medial are less investigated than interactions striato-cingulate, the authors should nevertheless acknowledge the importance of the later pathway.

The authors' response:

We now acknowledge the importance of the mPFC-to-VStr pathway in goal-directed behavior in the 2nd paragraph of page 7. As for possible differences between the ventral striatum and AM, it is hard to make a comparison. The level of ICSS largely depended on the expressions of opsins and the placements of stimulation probes in relation to the target regions. It is more difficult to achieve an optimal stimulation for the AM, because the AM is smaller than the ventral striatum. Some AM mice responded as vigorously as the best performer of ventral striatal mice, suggesting that motivational effect of AM stimulation may be as strong as that of ventral striatal stimulation.

In their last study the authors did conduct an fMRI study in humans. Observed effects are not located in their preferred ROI, IL, which would correspond to the subgenual cingulate cortex (area25) in humans (see for instance Schaeffer et al. PNAS 2020; Heilbronner et al. 2014 Biological Psychiatry). This discrepancy with results obtained in rodents might be related to the fact that the stronger effects were not in IL in rodents. Could the key area in the rodent studies be PL? Furthermore, the localized effect identified here speaks again for a functional heterogeneity of the medial wall.

The authors' response:

As discussed above, ROI was mPFC. We believe that it is important to distinguish between stimulation-induced stereotyped behavior and environmental event-induced adaptive behavior in considering apparent inconsistent results between rodents and humans. Although connectivity does not inform the nature of functional differences, topographical differences in connectivity suggest functional differences among the sub-regions of the mPFC. Indeed, a number of previously published studies showed differential roles of mPFC sub-regions. Particularly, the mPFC of humans is enlarged compared to that of rodents; enlarged mPFC may reflect extended repertoire of behavior that mPFC sub-areas regulate. However, we did not observe differences in the rate of ICSS between the IL and the rest of the mPFC. We suggest that the rate of ICSS is so stereotyped that it does not reflect the repertoire of behavior that mPFC sub-areas regulate. Instead, this measure may simply capture a common functional property of mPFC sub-regions. We now clearly state that we considered the mPFC as a unit in mouse experiments after the initial experiment (the 2nd paragraph on page 4) and discuss possible species differences in mPFC functions in the discussion section (the last paragraph on page 12, extending to the next page).

REVIEWER COMMENTS

Reviewer #1 (Remarks to the Author):

The authors have addressed all the concerns I raised in my original review. This is a very interesting paper that tackles a technologically challenging issue: demonstrating a specific role for the AM thalamic nucleus in mPFC dependent self-stimulation. I am confident that will have a significant impact on the field.

Reviewer #2 (Remarks to the Author):

The authors have done a thorough job in responding to my questions, comments, and concerns. They have included new data and analyses. I remain of the view that this an exciting and interesting manuscript.

Reviewer #3 (Remarks to the Author):

I am glad that the authors clarified many issues. They also conducted experiments suggesting that mPFC-AM circuit modulates motivation through its modulation of dopaminergic neuron activity. All my previous concerns have been well addressed and I am happy to recommend publication of this paper.

Reviewer #4 (Remarks to the Author):

I thank the authors for their answers to my queries and their revised and improved manuscript.

Authors' response to reviewers' comments

REVIEWERS' COMMENTS

Reviewer #1 (Remarks to the Author):

The authors have addressed all the concerns I raised in my original review. This is a very interesting paper that tackles a technologically challenging issue: demonstrating a specific role for the AM thalamic nucleus in mPFC dependent self-stimulation. I am confident that will have a significant impact on the field.

Reviewer #2 (Remarks to the Author):

The authors have done a thorough job in responding to my questions, comments, and concerns. They have included new data and analyses. I remain of the view that this an exciting and interesting manuscript.

Reviewer #3 (Remarks to the Author):

I am glad that the authors clarified many issues. They also conducted experiments suggesting that mPFC-AM circuit modulates motivation through its modulation of dopaminergic neuron activity. All my previous concerns have been well addressed and I am happy to recommend publication of this paper.

Reviewer #4 (Remarks to the Author):

I thank the authors for their answers to my queries and their revised and improved manuscript.

AUTHORS' RESPONSES

We thank all reviewers for their comments, which helped us to improve the paper.